# Exploring the Translation Mechanism of Large Language Models

**Hongbin Zhang[1,2]**, **Kehai Chen[1,2]**, **Xuefeng Bai[1]**, **Xiucheng Li[1]**, **Yang Xiang[2]**[*], **Min Zhang[1,2]**[*]

[1]*Institute of Computing and Intelligence, Harbin Institute of Technology, Shenzhen, China*
[2]*Peng Cheng Laboratory, Shenzhen, China*
*azure.starzhang@gmail.com, {chenkehai,baixuefeng,lixiucheng}@hit.edu.cn,*
*xiangy@pcl.ac.cn, zhangmin2021@hit.edu.cn*

## Abstract

While large language models (LLMs) demonstrate remarkable success in multilingual translation, their internal core translation mechanisms, even at the fundamental word level, remain insufficiently understood. To address this critical gap, this work introduces a systematic framework for interpreting the mechanism behind LLM translation from the perspective of computational components. This paper first proposes subspace-intervened path patching for precise, fine-grained causal analysis, enabling the detection of components crucial to translation tasks and subsequently characterizing their behavioral patterns in human-interpretable terms. Comprehensive experiments reveal that translation is predominantly driven by a sparse subset of components: specialized attention heads serve critical roles in extracting source language, translation indicators, and positional features, which are then integrated and processed by specific multi-layer perceptrons (MLPs) into intermediary English-centric latent representations before ultimately yielding the final translation. The significance of these findings is underscored by the empirical demonstration that targeted fine-tuning a minimal parameter subset ($< 5\%$) enhances translation performance while preserving general capabilities. This result further indicates that these crucial components generalize effectively to sentence-level translation and are instrumental in elucidating more intricate translation tasks. Code is available at this URL.

## 1 Introduction

Large language models (LLMs) have demonstrated strong capability to handle multilingual translation tasks (Zhu et al., 2024; Zhang et al., 2024; Gain et al., 2025), paving the way for a new paradigm in machine translation (Xu et al., 2024a; Alves et al., 2024) and progressively approaching human-level translation (Xu et al., 2024c; Lu et al., 2024; Xu et al., 2024b). Despite these successes, a comprehensive understanding of the internal core mechanisms underlying LLM translation is still lacking, even for the fundamental word-level translation. This significant gap in interpretability presents considerable challenges in ensuring reliability and further advancement in translation capability. Prior analyses concentrated on surface-level emergent linguistic phenomena (e.g., neuron activation patterns (Mu et al., 2024; Tang et al., 2024) or intermediate representations (Wendler et al., 2024; Zhu et al., 2024)), remaining *observational* rather than elucidating the *computational mechanistic basis* underlying translation. A comprehensive understanding of these functional mechanisms is critical for achieving robust improvements in translation capability and advancing the development of controllable and interpretable LLMs (Wang et al., 2023; Zhang et al., 2025).

---

[*]Corresponding Authors.

In this paper, we study the internal mechanism of LLM translation by progressively investigating the following research questions:

- *Which components of LLMs crucially contribute to performing translation?*
- *What behavioral patterns do these translation-crucial components exhibit?*
- *Can fine-tuning these translation-crucial components enhance LLM translation capability?*

To this end, this paper introduces a systematic framework that, by initially utilizing the proposed subspace-intervened path patching for precise, fine-grained causal analysis, examines the causal contributions of computational components to translation, thereby facilitating the detection of components crucial for translation tasks. Subsequently, for components judged as essential, we systematically analyze their behavioral patterns by (1) characterizing attention heads' specialized functional roles according to the attention contribution to lexical alignment and (2) measuring correlations between MLP representations and translation-relevant token embeddings.

Comprehensive analysis indicates that translation is predominantly driven by a sparse subset of attention heads, which can be characterized into three distinct functional roles: (i) *source heads* that focus on source-language tokens, (ii) *indicator heads* that track signals steering the translation task, and (iii) *positional heads* that maintain sequential coherence. Furthermore, we demonstrate that MLPs iteratively integrate translation-related features from these specialized attention heads, processing them into intermediate, English-centric latent representations.

Building on these insights, we design a targeted optimization strategy to selectively fine-tune translation-crucial components, thereby assessing whether this focused approach improves translation performance. We empirically find that such targeted fine-tuning of a minimal parameter subset enhances translation performance while preserving general capabilities, a finding that further underscores the effective generalization of these essential components to sentence-level translation.

In summary, our main findings are as follows:

- *Only a sparse subset of heads (less than 5%) are crucial for LLMs' translation.*
- *Crucial heads exhibit specialized functions by processing translation-relevant features, which MLPs then integrate and transform into English-centric latent representations.*
- *Fine-tuning merely 64 heads achieves performance parity with full-parameter fine-tuning.*

## 2 Related Works

**Neural machine translation interpretation.** Prior interpretability research in Neural Machine Translation (NMT) (Bau et al., 2019; Voita et al., 2019) has predominantly focused on sequence-to-sequence (seq2seq) models with encoder-decoder architectures, often analyzing individual attention head contributions via techniques like head pruning. To the best of our knowledge, this study is the first to investigate the translation mechanisms underlying decoder-only LLMs. Notably, our findings that translation is driven by a sparse subset of attention heads and attention heads play specialized functional roles align with previous research (Voita et al., 2019; Behnke and Heafield, 2020), suggesting the generalizability of these phenomena across different architectural designs.

**Mechanistic interpretability.** Mechanistic interpretability (MI) elucidates neural network mechanisms by seeking to reverse-engineer and decode their functioning (Meng et al., 2022; Lan et al., 2024; Zhao et al., 2024a; Rai et al., 2024). Path patching (Goldowsky-Dill et al., 2023; Wang et al., 2023), derived from activation patching (Heimersheim and Nanda, 2024; Zhang and Nanda, 2024), probes causal relationships and analyzes interactions between components in neural networks by tracing effect propagation along network pathways via targeted activation interventions. Recent studies highlight the utility of path patching to gain insights into functioning behavior, such as identifying circuits for tasks like indirect object identification (Wang et al., 2023) and arithmetic calculations (Zhang et al., 2025). To achieve a more precise and fine-grained causal analysis, this paper proposes subspace-intervened path patching to refine analytical precision and granularity.

**Interpretability in multilingual LLMs.** Recent studies have delved deeper into *how* LLMs achieve multilingualism by investigating linguistic phenomena emergent in multilingual context (Bhattacharya and Bojar, 2024; Peng and Søgaard, 2024; Ferrando and Costa-jussà, 2024; Dumas et al., 2024;

Zaranis et al., 2024). Key findings indicate that (i) increased linguistic diversity in inputs leads to reduced neuron activations (Mu et al., 2024); (ii) LLMs exhibit language-specific functional regions (Tang et al., 2024); and (iii) English frequently functions as an implicit computational pivot (Wendler et al., 2024; Zhao et al., 2024b). Unlike prior research centered on surface-level linguistic phenomena, this work comprehensively analyzes the underlying computational translation mechanisms in LLMs.

# 3   Constructing Analysis Dataset

To explore LLM translation mechanisms, we begin with word-level translation, which offers a more tractable, interpretable approach and provides a foundational first step to understanding core translation processes. Taking inspiration from the prompt design and word selection in Wendler et al. (2024), we construct a word translation dataset across five widely used languages (e.g., English (En), Chinese (Zh), Russian (Ru), German (De), and French (Fr)). Taking word translation from English to Chinese (En $\Rightarrow$ Zh) as an example, a word translation prompt containing the translation logic, such as "English: book - 中文: " ("中文" means "Chinese") might appear in the dataset. To eliminate task ambiguity and ensure a focused exploration of the translation mechanism, we select the samples that successfully prompting LLMs to translate, as positive data using the notation of $X_+$. More details of the construction of word translation datasets can be found in Appendix B.1.

For activation perturbation, we construct a negative dataset comprising counterfactual sentences that exclude translation logic, using the notation of $X_-$. The negative samples are generated adhering to two core principles: (1) preserving grammatical structures from the original $X_+$ sentences and (2) replacing several crucial words responsible for the translation logic with contextually irrelevant terms. For instance, a sentence from $X_+$ like "English: cloud - 中文: _" is replaced with the corresponding counterfactual one "English: cloud - Nothing: _". This isolates the model's impact on translation tasks from sentence structural or syntactic variables, enabling precise analysis of how LLMs perform translation tasks. The details of multiple counterfactual templates are provided in Appendix B.2.

# 4   Crucial Translation Components Detection

We begin by addressing the first research question: "*Which components crucially influence LLMs' translation capabilities?*" By leveraging the proposed subspace intervened path patching (§4.1), we detect components crucial for performing translation tasks (§4.2), subsequently validate their importance through knockout (§4.3), and further examine whether these components exhibit consistency across pre-training and post-training phases (§4.4).

## 4.1   Subspace Intervened Path Patching

Motivated by the linear representation hypothesis that linear subspaces of vectors will be the most interpretable model components (Geiger et al., 2024; Makelov et al., 2024; Park et al., 2024), this paper proposes subspace-intervened path patching. This method first identifies a "translation-steering" subspace within a component's activations using contrastive translation data pairs in an unsupervised manner. Subsequently, interventions are confined to the "translation-steering" subspace, enabling a precise analysis of the component's causal effect on the final translation.

**Identification of translation-steering subspace.** Building on previous work, which indicates that contrastive pairs are optimal choice for extracting desired behaviors from LLMs (Zou et al., 2025; Højer et al., 2025), the proposed method identifies a translation-steering subspace. This is achieved by utilizing translation contrastive activations—specifically, the difference in activations between input yielding correct translations and those lacking translation logic—to effectively capture the translation signal while excluding homogeneous noise. Formally, for an input sequence $x$, an activation vector $\mathbf{a}_c(x) \in \mathbb{R}^d$ is extracted from component $c$ at the final token position, where $d$ denotes the component's hidden dimension. A curated analysis dataset, comprising $N$ contrastive pairs $(X_+^{(i)}, X_-^{(i)})$ (details in §3), is utilized. For each pair $i$, the contrastive activation vector $\Delta\mathbf{a}_c^{(i)}$ is computed as the difference between the activations from the reference input $X_+^{(i)}$ and the counterfactual input $X_-^{(i)}$. To analyze dominant directions of activation shifts in the analysis dataset, these contrastive vectors, $\{\Delta\mathbf{a}_c^{(i)}\}_{i=1}^N$, form the columns of an activation difference matrix

$\boldsymbol{M}_c \in \mathbb{R}^{d \times N}$. Inspired by prior research (Xie et al., 2022; Makelov et al., 2024), we hypothesize that $\boldsymbol{M}_c$ can be decomposed into two orthogonal subspaces: (i) a universal translation-steering subspace $\boldsymbol{S}_c$, representing translation directions shared across word translation datasets, and (ii) a specific subspace $\boldsymbol{E}_c$, capturing dataset-specific features. Following the methodology of Xie et al. (2022); Piratla et al. (2020), this decomposition is achieved by optimizing the objective:

$$\min_{\boldsymbol{S}_c, \boldsymbol{E}_c, \boldsymbol{\Gamma}} ||\boldsymbol{M}_c - \boldsymbol{S}_c \mathbb{1}^\top - \boldsymbol{E}_c \boldsymbol{\Gamma}^\top||_F$$
$$\text{s.t.} \quad \text{Span}(\boldsymbol{S}_c) \perp \text{Span}(\boldsymbol{E}_c), \tag{1}$$

where $\boldsymbol{S}_c \in \mathbb{R}^{d \times 1}$, $\boldsymbol{E}_c \in \mathbb{R}^{d \times r}$, and $\boldsymbol{\Gamma} \in \mathbb{R}^{N \times r}$ contains the coordinates of the dataset-specific signals projected onto these $r$ components. Algorithm 1 presents the overall procedure to obtain $\boldsymbol{S}_c$.[2]

**Subspace projection patching.** Path patching (Wang et al., 2023; Zhang et al., 2025) traces influence from a *Sender* to a *Receiver* node. This involves replacing the activation of component $c$ from an original input, $\mathbf{a}_c(X_+)$, with its activation from a counterfactual input, $\mathbf{a}_c(X_-)$.[3] The proposed method refines this by confining the intervention to a pre-defined, task-steering subspace $\boldsymbol{S}_c$ within the activation space of component $c$. Formally, let $W_c \in \mathbb{R}^{d \times k}$ be a matrix whose columns form an orthonormal basis for $\boldsymbol{S}_c$. The orthogonal projection operator onto this subspace is $P_{\boldsymbol{S}_c} = W_c W_c^T$, and the projection onto its orthogonal complement $\boldsymbol{S}_c^\perp$ is $P_{\boldsymbol{S}_c^\perp} = I - P_{\boldsymbol{S}_c} = I - W_c W_c^T$. The patched activation, $\tilde{\mathbf{a}}_c$, is constructed by combining the projection of the counterfactual activation $\mathbf{a}_c(X_-^{(i)})$ onto $\boldsymbol{S}_c$ with the projection of the original activation $\mathbf{a}_c(X_+^{(i)})$ onto $\boldsymbol{S}_c^\perp$ (Equation 2):

---

**Algorithm 1** Task Steering Subspace Identification

**Require:** Set $\{(X_+^{(i)}, X_-^{(i)})\}_{i=1}^N$ of $N$ contrastive data pairs, rank of specific subspace $r$.
**Ensure:** Task-steering subspace $\boldsymbol{S}_c$
    *Phase 1: Compute contrastive activations*
1: **for** $i \leftarrow 1$ to $N$ **do**
2:     $\Delta \mathbf{a}_c^{(i)} \leftarrow \mathbf{a}_c(X_+^{(i)}) - \mathbf{a}_c(X_-^{(i)})$
3: **end for**
4: $\boldsymbol{M}_c \leftarrow \{\Delta \mathbf{a}_c^{(i)}\}_{i=1}^N$
    *Phase 2: Approximation of $\boldsymbol{M}_c$ with rank $r$*
5: $\boldsymbol{S}_c' \leftarrow \frac{1}{d} \boldsymbol{M}_c \mathbb{1}$
6: $\boldsymbol{E}_c', \_, \boldsymbol{\Gamma}' \leftarrow \text{Top-}r\text{-SVD}\left(\boldsymbol{M}_c - \boldsymbol{S}_c' \mathbb{1}^\top\right)$
7: $\boldsymbol{M}_c' \leftarrow \boldsymbol{S}_c' \mathbb{1}^\top + \boldsymbol{E}_c'(\boldsymbol{\Gamma}')^\top$
    *Phase 3: Force orthogonal constraint of objective 1*
8: $\boldsymbol{S}_c \leftarrow \frac{1}{\|(\boldsymbol{M}_c')^+ \mathbb{1}\|^2}(\boldsymbol{M}_c')^+ \mathbb{1}$
9: **return** $\boldsymbol{S}_c$

---

$$\tilde{\mathbf{a}}_c = P_{\boldsymbol{S}_c}\mathbf{a}_c(X_-^{(i)}) + P_{\boldsymbol{S}_c^\perp}\mathbf{a}_c(X_+^{(i)}) = W_c W_c^T \mathbf{a}_c(X_-^{(i)}) + (I - W_c W_c^T)\mathbf{a}_c(X_+^{(i)}). \tag{2}$$

The causal effect mediated by $\boldsymbol{S}_c$ along the Sender $\rightarrow$ Receiver path is measured by the changes in model output (e.g., the logit of ground-truth tokens), which mitigates confounding effects and yields more precise and interpretable estimates of causal contributions. Algorithm 2 shows the procedure.

## 4.2 Detection of Translation-Crucial Components

We then apply the proposed subspace-intervened path patching to precisely analyze the causal relationship between components and translation capability and detect the translation-critical components.

**Detection results of crucial heads.** This study examines the causal impact on output logits from path patching individual heads across layers in LLaMA2-7B (Touvron et al., 2023). Our analysis focuses on two translation directions: Chinese to other languages (Zh $\Rightarrow$ X) and vice-versa (X $\Rightarrow$ Zh)[4]. Following the criteria of previous work (Zhang et al., 2025), we define "crucial heads" as those whose magnitude of logit change exceeds $1.0\%$, a threshold empirically determined as most contributions fall within $\pm1.0\%$ and consistent with prior studies (Wang et al., 2023; Heimersheim and Nanda, 2024). As depicted in Figure 1, where each square at position $(x, y)$ denotes the $x$-th head in the $y$-th layer, several key findings emerge:

1. **Only a sparse subset of heads significantly influences translation performance.** For instance, if patching the head at position $(8, 31)$ results in a substantial decrease in the target token's logit value, illustrating its critical role in the translation process.

---

[2]The details and theoretical justification is are provided in Appendix C.

[3]Appendix D.1 provides details on standard path patching, while Appendix D.2 presents a comparation.

[4]For robustness, we also conduct additional experiments on detecting crucial components in other LLMs and other directions (e.g., En $\Rightarrow$ X, and X $\Rightarrow$ En). Details are provided in the Appendix G.

**Algorithm 2** Subspace Intervened Path Patching

**Require:** Set $\mathcal{D} = \{(X_+^{(i)}, X_-^{(i)})\}_{i=1}^N$ of $N$ contrastive data pairs, model $M$, set of model components $C$, set of task-steering subspace basis matrices $\{W_c \mid c \in C\}$.
**Ensure:** Node importance scores $\Delta = \{\delta_c \mid c \in C\}$.

1: **for** $k \leftarrow 1$ to $N$ **do**             $\triangleright$ Iterate over each data pair $(X_+^{(i)}, X_-^{(i)})$ in $\mathcal{D}$
2:  Compute base activations $\mathbf{a}(X_+^{(i)})$ and $\mathbf{a}(X_-^{(i)})$.
3:  $y_{\text{orig}}^{(i)} \leftarrow M(X_+^{(i)})$    $\triangleright$ Compute original model output (e.g., a specific logit) for $X_+^{(i)}$
4:  **for** each component $c \in C$ **do**        $\triangleright$ Iterate over each model component
5:    $\tilde{\mathbf{a}}^{(i)} \leftarrow \mathbf{a}(X_+^{(i)})$   $\triangleright$ Initialize the full hybrid activation set with reference activations $\mathbf{a}(X_+^{(i)})$
6:    $\tilde{\mathbf{a}}_c^{(i)} \leftarrow W_c W_c^{\mathrm{T}} \mathbf{a}_c(X_-^{(i)}) + (I - W_c W_c^{\mathrm{T}}) \mathbf{a}_c(X_+^{(i)})$ $\triangleright$ Equation 2 for subspace projection
7:    $y_{\text{new}}^{(i)} \leftarrow M(X_+^{(i)}; \tilde{\mathbf{a}}^{(i)})$   $\triangleright$ Compute model output using the hybrid activation set $\tilde{\mathbf{A}}^{(i)}$
8:    $\delta_c^{(i)} \leftarrow \frac{y_{\text{new}}^{(i)} - y_{\text{orig}}^{(i)}}{y_{\text{orig}}^{(i)} + \epsilon}$ $\triangleright$ Calculate the relative change in output due to patching component $c$
9:  **end for**
10: **end for**
11: **for** each component $c \in C$ **do**   $\triangleright$ Aggregate the effects for each component across datasets
12:  $\delta_c \leftarrow \frac{1}{N} \sum_{k=1}^N \delta_c^{(i)}$     $\triangleright$ Average the individual effects $\delta_c^{(i)}$ for component $c$
13: **end for**
14: **return** $\Delta$      $\triangleright$ Return the set of aggregated node/component importance scores

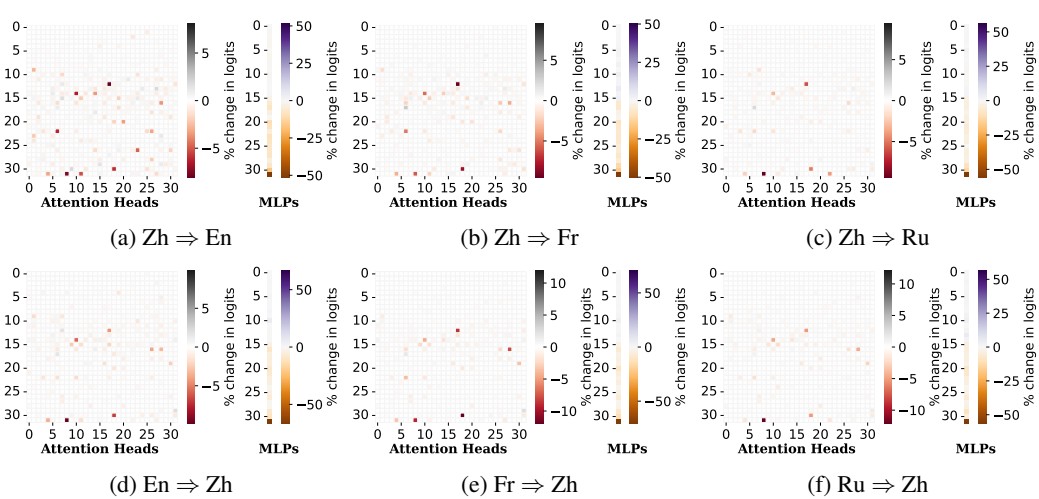

(a) Zh ⇒ En       (b) Zh ⇒ Fr       (c) Zh ⇒ Ru

(d) En ⇒ Zh       (e) Fr ⇒ Zh       (f) Ru ⇒ Zh

Figure 1: Importance of heads related to translation across different directions. Each square at position $(x, y)$ refers to the $x$-th head in the $y$-th layer. Red (Brown) squares denote heads (MLPs) that have a positive impact on predicting the target token, while grey (purple) squares indicate heads (MLPs) with a negative effect. Additional explanations of this figure are available in Apdx. D.3.

2. **Impactful heads are concentrated in the middle and final layers.** Earlier layers lack heads directly influencing target token logits; instead, crucial heads cluster predominantly between layers 12 and 20 and in the final two layers. This pattern remains consistent across all translation directions.

3. **Crucial heads exhibit high transferability across translation directions.** A notable finding is the significant overlap of crucial heads across diverse language pairs. Analysis reveals that language pairs sharing the same source or target language (e.g., En ⇒ Zh and Fr ⇒ Zh) exhibit a crucial attention head overlap exceeding 70%, while bidirectional translation pairs (e.g., Fr ⇔ En) surpass 60%. This overlap suggests these heads serve generalizable functions in translation, independent of translation directions. Their consistency across language pairs underscores their importance and transferability, indicating contributions to core translation mechanisms regardless of specific languages.

**Detection results of crucial MLPs.** Similar to crucial heads, most MLPs in earlier layers (0–14) exhibit negligible influence on output logits, with changes confined to approximately ±0.0%. Crucial

MLPs cluster predominantly after layer 15, exceeding $5.0\%$ logit change, whereas the final layer MLP exhibits a substantial impact—reaching $50.0\%$ on target token logit change. This strong correlation between later MLP layers and logit changes underscores their critical role in shaping translations.

**Extend mechanistic causal analysis to more settings.** To validate the generalization of the proposed subspace-intervened path patching, we extend mechanistic causal analysis to three additional settings:

1.**Low-resource and typologically diverse language pairs** (Swahili, Bengali, and Arabic) (Appendix E.1): Results presented in Table 11 and Figure 9 demonstrate that the sparsity and transferability of crucial attention heads still persist across low-resource and typologically diverse language pairs, substantiating these characteristics as fundamental translation mechanism of LLMs that are independent of resource availability or linguistic typology.

2.**Sentence-level translation** using the WMT23 English-to-Chinese dataset (Kocmi et al., 2023) (Appendix E.2): Causal analysis results in Table 12 revealed a 46.9% overlap between the top crucial attention heads for word-level and sentence-level translation tasks. Ablation experiments demonstrated that knocking out five shared heads resulted in significant performance degradation for both word-level (-39% in logits) and sentence-level (+36% in PPL) translation tasks, whereas ablating five heads crucial exclusively for sentence-level translation had minimal impact on word-level performance (-2%) but caused substantial degradation in sentence-level translation (+43% in PPL), highlighting the functional specialization of attention heads for sentence-level translation.

3.**Multilingual mathematical reasoning** using MGSM (Shi et al., 2023) (Appendix E.3): A key strength of the proposed subspace-intervened path patching is its task-agnostic ability to generalize across different tasks without requiring task-specific modifications. We then extend the mechanistic causal analysis to the multilingual mathematical reasoning task. We generated counterfactual examples by altering mathematical instructions while preserving the core mathematical content, following the procedure outlined in Section 3. Some examples are listed in Table 13. Our analysis revealed a sparse set of critical attention heads for mathematical reasoning, comprising only 3.95% of all heads in the model. This sparsity pattern aligns with our findings regarding the translation mechanism, demonstrating consistency across different cognitive tasks. Ablating the top-10 critical heads caused a 60% drop in reasoning accuracy, confirming their mathematical reasoning functional importance.

## 4.3 Validating Crucial Components Through Knockout

Interpretive analyses of model components risk misleading or non-rigorous (Bolukbasi et al., 2021; Wiegreffe and Pinter, 2019). To ensure reliability, we validate the significance of detected crucial components via *mean ablation* (Wang et al., 2023). This method replaces a component's activation with average activations across counterfactual data $X_-$, effectively neutralizing its task-specific information. Performance decline confirms a component's importance for translation tasks, whereas no significant performance change suggests it is not critical.

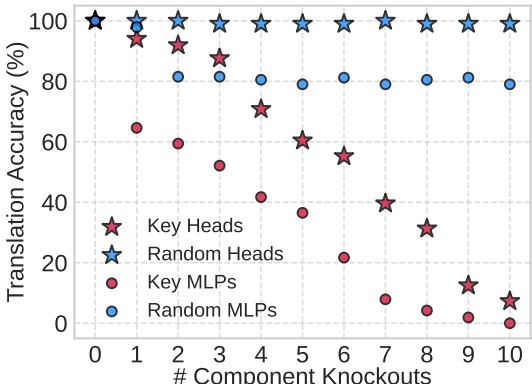

Figure 2: Translation accuracy changes when components are progressively knocked out.

**Validation results on the analysis dataset.** We examine how incrementally knocking out En $\Rightarrow$ Zh crucial heads affects LLM translation performance on the analysis dataset[5]. As shown in Figure 2, disabling "*crucial heads*" leads to a significant decline in translation accuracy, whereas knocking out "*random heads*" causes minor fluctuations, with accuracy remaining stable within 2%. A similar trend can be observed when knocking out MLPs. These results highlight the essential role of the detected key attention heads in sustaining the translation capability of LLM.

## 4.4 Examine Consistency of Crucial Components Across Training

---

[5]We have also conducted validation experiments on randomly selected datasets, see Appendix H.

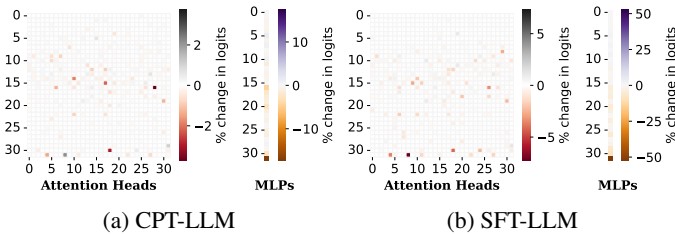

(a) CPT-LLM        (b) SFT-LLM

Figure 3: Importance of components related to **En⇒ Zh** translation across LLaMA-2-7B after CPT or SFT.

To investigate whether crucial attention heads remain consistent across distinct training phases, we analyze (1) continued pre-training (CPT) (Xu et al., 2024a) on the LLaMA-2-7B base model on 1 billion tokens of OSCAR data (Ortiz Suárez et al., 2020) and (2) supervised fine-tuning (SFT) (Jiao et al., 2023) on LLaMA-2-7B base model on the WMT17-22 validation dataset.

**Detection results.** As illustrated in Figure 3, compared to the base LLM results in Figure 1d, LLMs after CPT exhibit significant distributional shifts in translation-crucial heads, whereas changes are minimal after SFT. This finding is statistically supported by our Two-Sample Kolmogorov-Smirnov

Table 1: Statistical comparison of logit changes between base model and trained models

| Comparison | K-S Test p-value | # Changed Heads | Max $\Delta_{\text{logits}}$ |
|---|---|---|---|
| Base vs. SFT | 0.355 | 8 of 32 | 3.12 |
| Base vs. CPT | < 0.00001 | 17 of 32 | 12.03 |

test on overall logit change distributions (Table 1), which revealed that CPT induces a significant distributional shift ($p < 0.00001$) within the top 32 attention heads, while SFT does not ($p = 0.355$).

**Discussion of the emergence of translation capability.** A comparative causal analysis incorporating a randomly initialized baseline further elucidates these findings. The randomly initialized model exhibited no specialized translation heads, whereas the base pre-trained model developed critical translation heads with a statistically significant distributional shift from the random baseline. In contrast, the SFT model showed only a minor, non-significant distributional shift relative to the pre-trained model. These results demonstrate that the pre-training stage fundamentally alters LLMs' translation capabilities, while supervised fine-tuning primarily focuses on localized parameter adjustments without modifying core abilities. Additional details are provided in Appendix F.

## 5 Behavioral Patterns Analysis

Motivated by the sparse distribution of crucial heads, we now turn to the second research question: "*What behavioral patterns do translation-crucial components exhibit?*" by systematically investigating their computational mechanisms through two interpretable diagnostic methods: (1) visualizing attention patterns to characterize the roles of crucial heads (Section §5.1), and (2) projecting MLP representation to measure correlations with translation-related token embeddings (Section §5.2).

### 5.1 Analysis of Attention Head

Acknowledging that attention weights alone may not fully explain model behavior (Kobayashi et al., 2020), this study investigates attention outputs to analyze significant token interactions during translation. Formally, for each analyzed head $(i, j)$, its weighted value output, $\mathbf{O}^{(i,j)} \in \mathbb{R}^{N \times N}$, is defined as in Equation 3:

$$\mathbf{O}^{(i,j)} = ||\mathbf{A}^{(i,j)}(x\mathbf{W}_V^{(i,j)})||_F, \tag{3}$$

where $N$ represents the sequence length, $\mathbf{A}^{(i,j)} \in \mathbb{R}^{N \times N}$ contains the attention weights, $x \in \mathbb{R}^{N \times d_{\text{model}}}$ is the input sequence representation, $\mathbf{W}_V^{(i,j)} \in \mathbb{R}^{d_{\text{model}} \times d_{\text{head}}}$ is the value weight matrix, and $d_{\text{model}}, d_{\text{head}}$ are the hidden dimension of model and head respectively. The role of each head is then determined by analyzing the salient features of $\mathbf{O}_{\text{END},:}^{(i,j)} \in \mathbb{R}^{1 \times N}$, which represents the interaction between the Query token at the END position and all Key tokens.

**Characterizing heads.** We first gain an intuitive insight into the "behavioral pattern" of the translation-crucial heads by visualizing attention values[6] as shown in the case in Figure 4. Building on the distinct focus patterns these heads exhibit across different input token types, and following Voita et al. (2019), we further categorize them into three distinct functional roles (illustrative examples are provided in Appendix B.3):

---

[6]Focus on Zh ⇒ En, with more directions results seen Appendix J.

1) **Source Heads** demonstrate concentrated attention on source-language tokens, specializing in cross-lingual alignment. These heads facilitate lexical transfer by identifying source language tokens among the input sequence.

2) **Indicator Heads** exhibit spike-shaped attention patterns on translation-specific indicators (e.g., language identifiers like "English" or "中文", and structural cues like colons), assisting translation mode recognition and syntactic boundary detection.

3) **Positional Heads** predominantly attend to adjacent tokens, managing contextual dependencies and resolving grammatical agreement.

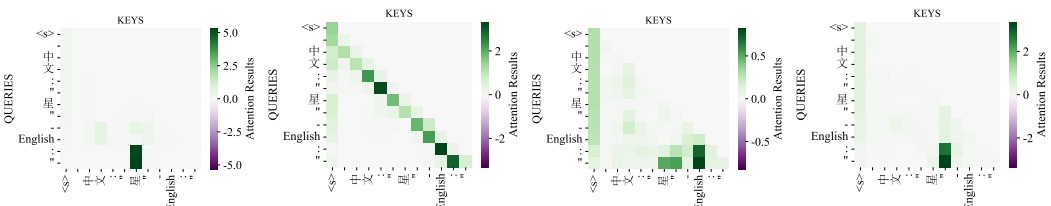

(a) Source Head $(18, 17)$    (b) Positional Head $(4, 31)$    (c) Indicator Head $(27, 14)$    (d) Indicator Head $(4, 14)$

Figure 4: The attention values visualization of the role-classified key heads in Zh $\Rightarrow$ En, which show different characteristics of different crucial heads.

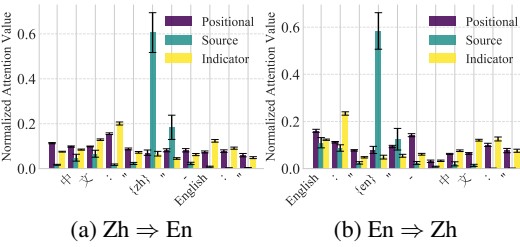

(a) Zh $\Rightarrow$ En       (b) En $\Rightarrow$ Zh

Figure 5: Mean and standard deviation of attention values from key head roles across input tokens.

**Distinct attention distribution across heads.** To quantitatively analyze the distinct patterns of heads' roles, we plot the distribution of their attention values on 100 randomly selected samples for Zh $\Leftrightarrow$ En translation tasks[7]. Figure 5 demonstrates that source heads predominantly focus on source tokens, positional heads distribute attention uniformly across the input context, and indicator heads concentrate on translation task indicator tokens, with all types showing minimal attention to irrelevant tokens.

## 5.2 Analysis of MLP

This study analyzes the linguistic content encoded in the inputs ($MLP_{in}$) and outputs ($MLP_{out}$) of MLP layers, focusing on translation-steering tokens: the translation indicator (IND), source language (SRC), and target language (TGT). To achieve this, we utilize the unembedding matrix $W_U \in \mathbb{R}^{d_{\text{model}} \times |\mathcal{V}|}$ (i.e., the final linear layer that projects hidden states of dimension $d_{\text{modal}}$ onto the vocabulary space of size $|\mathcal{V}|$) as a diagnostic probe, where $W_U[\text{TOK}]$ denotes the unembedding vector corresponding to a specific token TOK. To quantify linguistic information propagation through MLP layers, we compute cosine similarities, denoted as $\langle MLP, \text{TOK} \rangle$, between $W_U[\text{TOK}]$ and both $MLP_{in}$ and $MLP_{out}$. Furthermore, to isolate the MLP layer's specific contribution, we follow Geva et al. (2022) by evaluating the cosine similarity of the layer's normalized change vector ($MLP_{out} - MLP_{in}$) with the normalized token embedding, as defined in Equation 4:

$$\langle MLP_{out} - MLP_{in}, \text{TOK} \rangle = \frac{MLP_{out} - MLP_{in}}{\|MLP_{out} - MLP_{in}\|} \cdot \frac{W_U[\text{TOK}]}{\|W_U[\text{TOK}]\|}. \tag{4}$$

**MLPs iteratively process translation-related features to generate target translations.** Analysis of MLP interactions with source and target tokens in 100 En $\Rightarrow$ Zh samples (Figure 6) reveals distinct operational phases across layers. Initially (layers 1–14), Figure 6a shows $\langle MLP_{in}, \text{SRC} \rangle$ values remain near-zero, indicating minimal source token encoding, consistent with the inactive region before layer 14 (Figure 1d). A significant increase in $\langle MLP_{in}, \text{SRC} \rangle$ occurs between layers 15–25, correlating with the activation of key attention heads, as source information is encoded in MLP representation. Subsequently, from layers 25–31, $\langle MLP_{in}, \text{SRC} \rangle$ decreases, signaling a transition towards target translation. Concurrently, ($\langle MLP_{in}, \text{IND} \rangle$) begin to rise after layer 12, peaking in

---

[7]Statistical significance analysis is available in Appendix I

the final layers to facilitate coherent target-language generation. Control comparisons using random English tokens ($\langle MLP_{in}, \text{RAND}\rangle$) consistently remain near-zero, confirming the observed pattern's specificity. Furthermore, Figure 6b demonstrates that from layer 15, where MLPs begin processing target token information, $\langle MLP_{out} - MLP_{in}, W_U[\text{TARGET}]\rangle$ progressively increases, suggesting the generation of translation. The phenomenon is generalizable as evidenced by similar trends in other LLMs (Appendix J).

**MLP intermediate features reveal an English-centric latent representation as a translational intermediary.** Further investigation into the correlation between MLP intermediate representations and the unembedding vector of semantically equivalent tokens across different languages during non-English translation pairs (e.g., De/Ru $\Rightarrow$ Zh) yields a significant finding. As illustrated in Figure 7, the similarity of these intermediate representations to English unembedding vectors is markedly higher in layers 16-26 compared to other languages, subsequently declining in layers 25-31. This pattern strongly suggests that LLM employs a "bridge-translation" mechanism. In this process, source inputs appear to be processed into an English-centric latent space before generating target language outputs, analogous to humans using their native language as a mental intermediary. This observation corroborates prior research (Wendler et al., 2024; Zhao et al., 2024b), affirming the pivotal role of English as a latent intermediary in multilingual LLM operations.

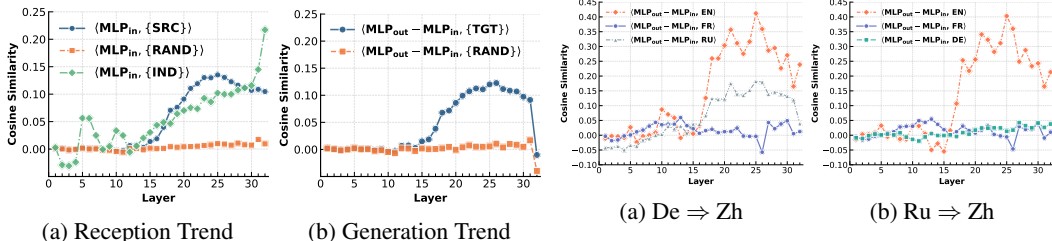

(a) Reception Trend    (b) Generation Trend      (a) De $\Rightarrow$ Zh     (b) Ru $\Rightarrow$ Zh

Figure 6: The correlation between MLP input or output with translation-related tokens.

Figure 7: The correlation between the MLP representation and the language's unembedding vector.

**More discussions on English as the pivot language.** Appendix M.1 presents a correlation analysis demonstrating the direct and significant impact of English-centricity on translation performance. We then explore different aspects of English as a pivot language: Appendix M.2 examines why English emerges as the pivot, while Appendix M.3 analyzes the role of English in forming the forward process of language models. Finally, Appendix M.4 investigates potential gender/formality translation biases introduced by English as a pivot latent representation.

# 6 Targeted Enhancement of Translation Capability

Building on the insights from two previous investigations, we aim to answer the final question: "*Can fine-tuning these translation-crucial components enhance LLM translation capability?*" We initiate by introducing our comparative experimental setup and results (Section §6.1) and further carry out two sets of analysis experiments (Section §6.2, and §6.3).

## 6.1 Experimental Setup and Results

**Experimental setup.** We examine three approaches on Zh $\Leftrightarrow$ En and De $\Leftrightarrow$ En directions: (1) full-parameter fine-tuning (Full SFT), (2) the proposed selectively fine-tuning of translation-crucial components (Targeted SFT), and (3) random-component fine-tuning (Random SFT), where random components match the parameter count of Targeted SFT. For training, we leverage human-parallel corpora (WMT17–WMT22, Flores-200 (Guzmán et al., 2019)) following Xu et al. (2024a), evaluating translation accuracy on WMT23/24 and general-domain benchmarks (MMLU (Hendrycks et al., 2021), ARC (Clark et al., 2018), SIQA (Sap et al., 2019)). More details are provided in Appendix K.

**Experimental results.** Tables 2 highlight three key advantages of Targeted SFT: (1) **Improved Translation Performance**: Targeted SFT significantly enhances translation performance across all language directions, surpassing Full SFT and substantially outperforming Random SFT. (2) **Preservation of General Capabilities**: Unlike Full SFT, which degrades performance on non-translation tasks, Targeted SFT maintains baseline general capabilities. (3) **Enhanced Training Efficiency**: It modifies fewer than 5% of parameters and reduces training time by half compared

Table 2: The overall evaluation results on Zh ⇔ En, De ⇔ En translation tasks and generic tasks.

| Models | Train Speed | Tuned Params. | Translation Tasks | | | | Generic Tasks | |
| | | | Zh⇒En | En⇒Zh | De⇒En | En⇒De | MMLU | Commonsense Reasoning |
| | | | BLEU↑/COMET↑/BLEURT↑ | | | | Acc. | Acc. |
|---|---|---|---|---|---|---|---|---|
| LLaMA2-7B | - | - | 15.6/73.1/56.6 | 17.0/74.1/55.9 | 24.8/76.8/62.1 | 13.0/64.2/49.1 | 45.9 | 55.3 |
| + Full SFT | 17sam./sec. | 6.7B | 20.4/78.7/63.9 | 30.3/80.7/62.9 | 35.4/83.4/70.7 | 27.9/78.3/63.7 | 42.6 | 50.2 |
| + Targeted SFT | 33sam./sec. | 0.27B | 21.3/79.1/64.3 | 30.7/81.4/64.3 | 37.1/83.7/71.4 | 27.6/78.4/63.8 | 46.0 | 55.7 |
| + Random SFT | 33sam./sec. | 0.27B | 16.9/76.9/61.1 | 26.4/79.3/61.6 | 32.5/81.6/68.1 | 22.7/76.2/60.3 | 45.9 | 54.9 |

to Full SFT. Furthermore, these positive results demonstrate that the detected translation-crucial heads generalize beyond isolated word translation and are significant to sentence-level translation. Additional results on more directions and LLMs are provided in Appendix L.

## 6.2 Language Transfer Evaluation of Crucial Translation Heads

This section evaluates the language transfer capabilities of crucial translation heads. Specifically, heads detected crucial for En ⇒ Zh translation were selected for fine-tuning and evaluating on En ⇔ Ja/Cs translation tasks. The comparable results, shown in Table 3, indicate that these translation-crucial attention heads exhibit cross-lingual generalization.

Table 3: The transfer evaluation results of En ⇒ Zh crucial heads on En ⇔ Cs and En ⇔ Ja.

| Models | En⇒Cs | En⇒Ja | Cs⇒En | Ja⇒En |
| | BLEU↑/COMET↑/BLEURT↑ | | | |
|---|---|---|---|---|
| LLaMA2-7B | 4.4/63.6/39.7 | 6.1/73.3/47.4 | 23.7/77.9/65.1 | 10.8/72.9/56.6 |
| + Full SFT | 20.2/80.0/66.5 | 15.2/82.4/56.7 | 31.9/83.1/71.7 | 17.4/79.5/64.1 |
| + Targeted SFT | 20.8/80.3/66.7 | 15.3/81.9/56.7 | 33.5/83.5/72.3 | 18.7/80.0/64.7 |
| + Random SFT | 15.8/78.5/63.8 | 11.3/79.9/53.7 | 29.1/81.5/68.8 | 14.0/77.9/62.1 |

Table 4: Ablative experiments on attention heads.

| Ablating Attention Heads | Train Speed | Tuned Params. | Zh ⇒ En | MMLU |
| | | | BLEU/COMET/BLEURT | Acc. |
|---|---|---|---|---|
| top-8 heads | 58sam./sec. | 0.017B | 18.7/78.1/63.0 | 46.1 |
| top-16 heads | 52sam./sec. | 0.033B | 20.0/78.4/63.5 | 45.9 |
| top-32 heads | 50sam./sec. | 0.067B | 20.4/78.6/63.8 | 45.8 |
| top-64 heads | 40sam./sec. | 0.134B | 21.3/79.1/64.3 | 45.9 |
| top-96 heads | 36sam./sec. | 0.134B | 21.0/79.0/64.2 | 45.7 |
| top-128 heads | 33sam./sec. | 0.268B | 21.1/79.1/64.4 | 45.5 |
| top-160 heads | 30sam./sec. | 0.335B | 21.3/79.1/64.4 | 45.3 |

Table 5: Ablative experiments on MLPs.

| Ablating MLPs | Train Speed | Tuned Params. | Zh ⇒ En | MMLU |
| | | | BLEU/COMET/BLEURT | Acc. |
|---|---|---|---|---|
| Top-64 heads | 33sam./sec. | 0.27B | 21.3/79.1/64.3 | 45.8 |
| +top-1 MLP | 30sam./sec. | 0.41B | 21.8/79.1/64.5 | 45.7 |
| +top-2 MLP | 27sam./sec. | 0.54B | 21.8/79.1/64.5 | 45.6 |
| +top-3 MLP | 24sam./sec. | 0.68B | 21.9/79.1/64.5 | 45.3 |
| +top-5 MLP | 20sam./sec. | 0.95B | 22.1/79.2/64.6 | 44.2 |
| +all MLP | 18sam./sec. | 4.62B | 22.5/79.4/64.7 | 42.8 |

## 6.3 Ablation Study of Trainable Components

Ablation studies on Zh ⇒ En translation were conducted to assess the impact of varying the number of fine-tuned attention heads and MLPs on translation performance, generic capabilities, and training efficiency. As indicated in Table 4, increasing the quantity of fine-tunable attention heads enhanced translation performance but concurrently weakened generic capabilities. Notably, fine-tuning 64 attention heads achieved an optimal balance between performance and computational cost. Furthermore, Table 5 reveals that while augmenting the number of MLPs improved translation performance, this approach more substantially degraded generic capabilities and reduced training speed compared to the fine-tuning of additional attention heads.

## 6.4 Supplementary Experiments

This section presents three additional experiments of targeted SFT: (1) evaluation results on domain-adaptive translation (Appendix N.1), (2) analysis of potential cultural bias amplification (Appendix N.2), and (3) qualitative case studies examining characteristic patterns (Appendix N.3).

## 7 Conclusion

This study systematically explores the translation mechanisms of LLMs by progressively addressing three research questions. We first identify components crucial for translation using our proposed subspace-intervened path patching, revealing that only a sparse subset of components (less than 5%) are indispensable. These heads exhibit specialized functions, extracting translation-related features, while MLPs integrate and process information towards intermediate, English-centric latent representations. Based on these findings, we empirically demonstrate that targeted fine-tuning of merely 64 translation-crucial heads achieves performance parity with full-parameter tuning. These results further emphasize the effectiveness of generalizing the detected crucial components to sentence-level translation. This work serves as a preliminary exploration of the translation mechanism underlying LLMs, establishing a solid foundation for elucidating more intricate translation tasks.

## Acknowledgment

We express our sincere gratitude to the reviewers for their valuable and insightful comments, which have significantly improved the quality of this work. This work was supported in part by the National Natural Science Foundation of China under Grant 62276077, Grant 62406091, and Grant U24A20328, in part by the Guangdong Basic and Applied Basic Research Foundation under Grant 2024A1515011205, in part by the Shenzhen College Stability Support Plan under Grant GXWD20231130104007001, in part by the Major Key Project of PCL under Grant PCL2025A12, and in part by the Shenzhen Science and Technology Program under Grant KQTD20240729102154066 and Grant ZDSYS20230626091203008.

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

# A Limitations and Discussion

**Limitations.** This study acknowledges a methodological consideration that guides future research directions. Although our parameter-aware methodology proves effective across open-source architectures, its applicability to closed-source systems remains theoretically constrained—a limitation that simultaneously highlights the urgent need for developing model-agnostic analysis frameworks in this evolving research domain.

**Potential impact.** This study pioneers the exploration of translation mechanisms at a fine-grained level by directly investigating the causal relationship between model components and translation performance. The employed interpretability techniques, such as attention visualization, distribution analysis, and unembedding quantification, are generalizable and can be extended to future research questions in interpretable machine learning. Furthermore, the systematic interpretability methodology presented is adaptable to other Natural Language Processing (NLP) tasks (e.g., summarization, question answering) and potentially to non-NLP domains, thereby encouraging further investigation into task-specific component analysis. The identification of universal translation components across diverse language pairs can inform the development of more robust multilingual Large Language Models (LLMs), particularly benefiting low-resource languages.

**Practical applications.** Practical applications of this study stemming from these insights are significant. Targeted fine-tuning, guided by the identification of key components, promises considerable computational efficiency. Specifically, the findings suggest that fine-tuning only essential components, rather than retraining entire models, can significantly reduce computational costs while preserving translation quality. Moreover, this research contributes to interpretable Artificial Intelligence (AI) for translation by offering a transparent, component-level understanding of how translation decisions are formulated. Such transparency is crucial for fostering trust and facilitating adoption in critical real-world scenarios, including legal, medical, and diplomatic applications.

**Future research.** Future research directions are also illuminated by this work. While the current analysis concentrated on word-level translation to isolate core mechanisms, subsequent studies could extend these insights to sentence-level and document-level contexts to achieve a more comprehensive understanding. Additionally, although this study focuses on specific components, the principles and findings can inform the design and analysis of larger and more complex models. As LLMs continue to increase in scale and complexity, a thorough understanding of their internal mechanisms becomes increasingly essential, and this work provides a foundational basis for such endeavors.

# B Translation Task Templates and Examples

As a clear case study, we first focus on Chinese due to its prevalence of single-token words and lack of spacing. We analyze Llama-2's vocabulary to identify single-token Chinese words (primarily nouns) with direct single-token English translations. This enables direct comparison of the model's next-token probabilities for correct Chinese words and their English equivalents. For robustness, we replicate experiments in German, Russian, and French, compiling datasets of 139 Chinese, 120 German, 115 Russian, and 118 French words.

## B.1 Dataset Construction

To ensure the next token is unambiguously inferable as a single token, we design translation prompts where $x_{n+1}$ is uniquely determined by the preceding context $x_1...x_n$. Each prompt specifies the source language, word, and target language, requiring the model to predict the translated word. Taking English-to-Chinese as an example, a word translation like "English: flower - 中文: 花" ("中文" means "Chinese", "花" means "flower") might naturally appear in the pretraining corpus. Such prompts explicitly guide Llama-2 to perform translation by leveraging its pretrained linguistic knowledge.

## B.2 Templates

We formalize counterfactual prompt generation through systematic grammatical preservation and semantic disruption, operating under two core design principles:

- **Structural Isomorphism**: Maintain original syntactic patterns (interrogative formats, place-holder positions, punctuation) while altering semantic content

- **Targeted Lexical Substitution**: Replace critical components through four operation classes

**Perturbation Taxonomy**    The perturbation strategies fall into four principal categories, as detailed in Table 6:

Table 6: Taxonomy of Counterfactual Perturbation Operations

| Operation Type | Implementation Mechanism |
|---|---|
| Target Nullification | Replace language identifiers with non-linguistic concepts (`{tgt_lang}` → "Void"/"Null") |
| Action Distortion | Substitute translation verbs with irrelevant actions ("translate" → "eat"/"delete") |
| Semantic Obfuscation | Alter task-specific nouns to disrupt functionality ("translation" → "color"/"flavor") |
| Paradox Insertion | Introduce self-contradictory modifiers ("into `{tgt_lang}`" → "into a silent rock") |

**Validation Protocol**    The constructed templates undergo rigorous verification:

1. *Grammatical Integrity Check*: Measure template fluency via language model perplexity scores (threshold: ≤15% deviation from originals)

2. *Task Disruption Test*: Verify semantic shift through human annotation (success criterion: ≥90% agreement on functionality removal)

The counterfactual prompts we used are shown in Table 7

Table 7: Examples of some regular translation prompt templates and counterfactual prompt templates.

| Normal Prompt | Counterfactual Prompt | Perturbation Type |
|---|---|---|
| `{src_lang}: "{src_word}" - {tgt_lang}: "{tgt_word}` | `{src_lang}: "{src_word}" - There is nothing: "{tgt_word}` | Target Nullification |
| `Translate "{src_word}" into {tgt_lang}: "` | `Translate "{src_word}" into Nothing: "` | Target Nullification |
| `Translate the {src_lang} word "{src_word}" to {tgt_lang}: "` | `Translate the {src_lang} word "{src_word}" to Null: "` | Target Nullification |
| `From {src_lang}: "{src_word}" to {tgt_lang}: "` | `From {src_lang}: "{src_word}" to Nowhere: "` | Target Nullification |
| `Provide the translation of "{src_word}" from {src_lang} to {tgt_lang}: "` | `Provide the color of "{src_word}" from {src_lang} to {tgt_lang}: "` | Action Distortion |
| `Q: How do you say "{src_word}" in {tgt_lang}? A: "` | `Q: How do you eat "{src_word}" in {tgt_lang}? A: "` | Action Distortion |
| `Q: What is the {tgt_lang} translation "{src_word}"? A: "` | `Q: What is the {tgt_lang} flavor "{src_word}"? A: "` | Semantic Obfuscation |
| `Translate "{src_word}" into {tgt_lang}: "` | `Translate "{src_word}" into a silent rock: "` | Paradox Insertion |
| `Q: What is "{src_word}" translated into {tgt_lang}? A: "` | `Q: What is "{src_word}" erased into {tgt_lang}? A: "` | Action Distortion |
| `From {src_lang}: "{src_word}" - {tgt_lang}: "{tgt_word}` | `From {src_lang}: "{src_word}" - Disabled: "{tgt_word}` | Action Distortion |

Note: All placeholders ({src_lang}, {src_word}, etc.) follow actual implementation syntax. Counterfactual perturbations preserve original grammatical structures while altering translation semantics through targeted substitutions.

**Evidence supporting the choice of the contrastive template.** To further substantiate this choice, we present two key evidences of why this contrastive template is suitable:

- **Empirical Validation:** Applying the contrastive template consistently results in 0% accuracy, confirming that the template reliably triggers LLM not to perform translations.
- **Reference to Prior Work:** We drew inspiration from Wang et al. (2023), where manually created contrastive samples were used for the Indirect Object Identification (IOI) task. For example:
  - Original prompt: *The store Cody and Scott went to had a snack. Cody gave it to Scott.*
  - Contrastive prompt: *The store Cody and Andrew went to had a snack. Cody gave it to Scott.*

This approach ensures that by replacing a key entity (here, the indirect object), the resulting label is guaranteed to be incorrect. Similarly, in our translation task, replacing the target language indicator "English" with an irrelevant term such as "Nothing" ensures that the model deviates from the correct translation.

### B.3 Token Type Definitions and Examples

- **IND (Instructional Tokens)**: Structural or framing tokens that establish the translation context but are not part of the source content. These tokens provide necessary formatting or linguistic direction without contributing to the semantic content being translated.
- **SRC (Source Tokens)**: The actual input text intended for translation. These tokens represent the semantic content that needs to be converted from the source language to the target language.
- **TGT (Target Tokens)**: The translated output tokens in the target language. These represent the model's generated translation of the source content.

**Illustrative example:** To demonstrate this token classification, consider the translation sequence:

$$\texttt{English: cloud - 中文: 云}$$

The token type decomposition for this sequence is as follows: This classification scheme enables

| Token Type | Tokens | Functional Role |
|---|---|---|
| IND | "English", ":", "-", "中文" | Structural framing for translation context |
| SRC | "cloud" | Source content for translation |
| TGT | "云" | Translated output in target language |

Table 8: Token type classification for the example sequence

precise analysis of how different token types influence attention mechanisms and translation behavior in neural machine translation models. The IND tokens establish the translation framework, SRC tokens provide the semantic input, and TGT tokens represent the model's generated output, allowing for systematic examination of cross-lingual transfer patterns.

## C  Task Steering Subspace Probing

Inspired by prior research (Xie et al., 2022; Makelov et al., 2024), we hypothesize that the space $M_c$ can be decomposed into two orthogonal subspaces: (i) a universal translation-steering subspace $S_c$, embodying translation features common across word translation datasets, and (ii) a specific subspace $E_c$, isolating features unique to individual datasets. This decomposition is achieved by optimizing the objective outlined in Equation 1, following the methodology of Xie et al. (2022); Piratla et al. (2020). We anticipate a lower dimensionality for the universal subspace $S_c$ because it represents shared, fundamental patterns; such commonalities can inherently be captured by a more parsimonious set of basis vectors, leading to a compact representation. Conversely, the specific subspace $E_c$ is expected

to possess a higher dimensionality to effectively accommodate the diverse and distinct characteristics particular to each dataset or sample, which necessitate a richer representational capacity to capture their unique signals.

The optimal solution to Equation 1 is efficiently computed via Singular Value Decomposition (SVD), with the detailed procedure outlined in Algorithm 1. Theorem 1, presented in this section, provides the formal basis for this optimal solution. A comprehensive proof can be found in Piratla et al. (2020); Xie et al. (2022).

**Theorem 1.** *For any matrix $\boldsymbol{M_c} \in \mathbb{R}^{d \times N}$, Algorithm 1 returns matrices $\boldsymbol{S_c} \in \mathbb{R}^{d \times 1}$, $\boldsymbol{E_c} \in \mathbb{R}^{d \times r}$, and $\boldsymbol{\Gamma} \in \mathbb{R}^{N \times r}$ that minimize Equation 1 subject to the constraint $Span\left(\boldsymbol{S_c}\right) \perp Span\left(\boldsymbol{E_c}\right)$.*

# D   More details related to Path Patching

## D.1   Standard Path Patching

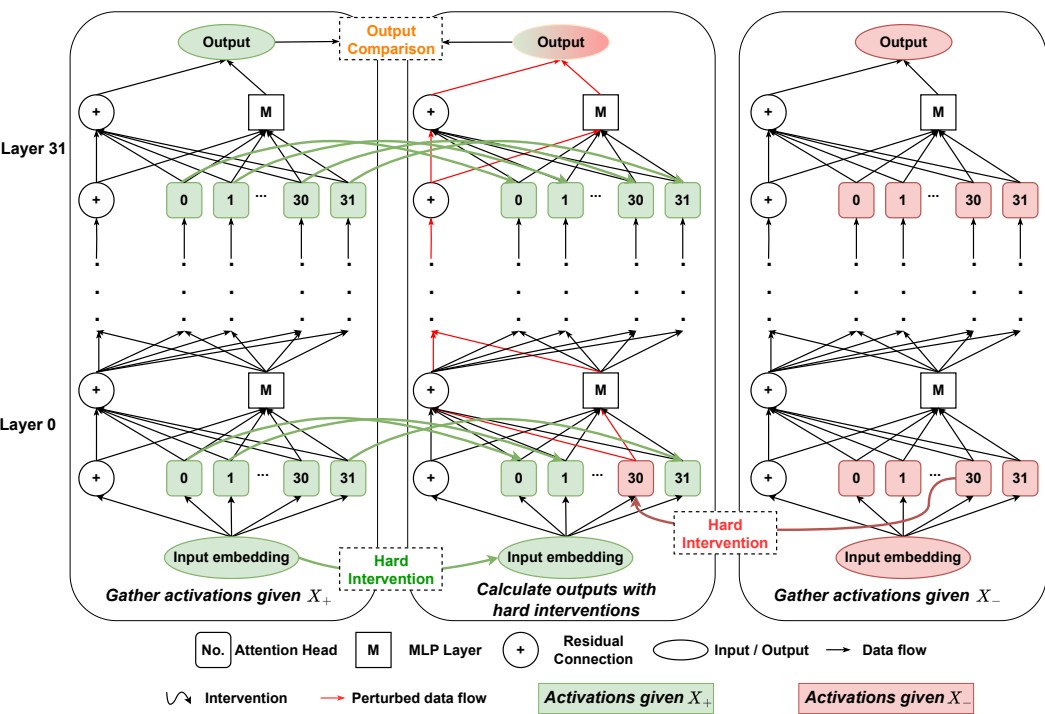

Figure 8: Illustration of the method "path patching". It measures the importance of the selected circuit (*i.e.*, the red lines that originate from Head 30 in Layer 0 to Output) for the transformer in completing the task on reference data.

The computation of large language models (LLMs) can be formalized as a directed acyclic graph (DAG) (Wang et al., 2023), where nodes represent computational components (e.g., attention heads, MLP layers) and edges denote directional data flow between them. Mechanistic interpretability seeks to reverse-engineer neural networks into interpretable algorithms, leveraging computational circuits as a framework. A computational circuit is a subgraph of the model's computational graph $M$, comprising nodes (e.g., embeddings, attention heads) and edges (e.g., residual connections, projections) that collectively implement specific tasks, such as translation.

The procedure of standard path patching is illustrated in Figure 8. Activations from all nodes are first recorded. A hard intervention replaces the Sender's activations with those from $X_-$, propagating the effect through paths $\mathcal{P}$ (residual connections and MLPs). Concurrently, other attention heads are frozen to $X_+$ to isolate the Sender's impact. The resulting logits are compared to quantify the Sender's causal contribution: significant changes indicate critical paths for task execution. Since residual streams and MLPs process tokens independently (Elhage et al., 2021), perturbing activations at the END token position suffices to measure effects on next-token prediction.

## D.2 Comparison of the proposed subspace-intervened path patching with standard path patching

Standard path patching techniques intervene on the entire activation vector of a component within neural networks (Heimersheim and Nanda, 2024; Wang et al., 2023). However, these activations often exhibit polysemanticity, simultaneously encoding multiple unrelated concepts. This polysemantic nature presents a significant challenge in mechanistic interpretability, as full-vector patching conflates the causal effects of target functions (such as translation) with numerous irrelevant functions encoded within the same vector space.

To address this limitation, our proposed method identifies and intervenes upon low-dimensional subspaces specifically responsible for translation within the activation space. This subspace-intervened approach enables the isolation of specific causal mechanisms of translation from confounding functionalities, providing a more fine-grained and accurate understanding of the model's internal translation processes. By moving from full-vector to subspace intervention, we achieve a targeted and necessary design that facilitates precise mechanistic analysis in large language models.

To validate the effectiveness of our subspace-intervened path patching approach, we conducted comprehensive experiments comparing it with standard path patching baselines across multiple translation directions. Our evaluation encompassed both high-resource (English-Chinese) and low-resource (English-Swahili) language pairs to assess the generalizability of our method.

We implemented both approaches on the same pre-trained multilingual language model architecture. For standard path patching, we followed the methodology described in prior work, intervening on complete activation vectors. For our subspace-intervened approach, we first identified translation-specific subspaces through targeted projection techniques before performing interventions.

Evaluation metrics included:

- Average logit changes when intervening on identified components
- Accuracy drop when knocking out the top-5 most crucial attention heads
- Translation performance measured by BLEU, COMET, and BLEURT scores after targeted supervised fine-tuning (SFT) of the top-32 identified heads

Table 9 presents a detailed comparison between standard path patching and our subspace-intervened approach across multiple translation directions. The results demonstrate the superior performance of our method in identifying components critical to translation.

Table 9: Comparison of standard path patching versus subspace-intervened approach across translation directions

| Translation Pairs | Top Crucial Heads Layer, Head | Avg. Logits Change | Acc. Drop Knockout Top-5 | Targeted SFT Performance BLEU/COMET/BLEURT |
|---|---|---|---|---|
| En→Zh (standard) | (31, 8), (14, 10), (30, 18) | -2.69% | -25% | 27.3/79.8/62.4 |
| En→Zh (subspace) | (15, 21), (31, 11), (18, 26) | -4.47% | -39% | 28.9/80.5/63.1 |
| Zh→En (standard) | (15, 19), (31, 22), (14, 10) | -1.71% | -22% | 18.5/77.9/62.8 |
| Zh→En (subspace) | (31, 27), (31, 11), (14, 14) | -2.49% | -31% | 19.8/78.4/63.3 |
| En→Sw (standard) | (22, 17), (31, 8), (16, 6) | -3.12% | -28% | 1.83/51.5/40.9 |
| En→Sw (subspace) | (16, 26), (31, 8), (18, 11) | -6.81% | -42% | 3.91/55.1/43.7 |
| Sw→En (standard) | (14, 14), (31, 22), (15, 11) | -1.43% | -21% | 14.5/67.1/53.2 |
| Sw→En (subspace) | (31, 27), (30, 18), (14, 10) | -2.01% | -26% | 15.9/67.9/54.0 |

The results reveal several key findings. First, our subspace-intervened method identifies components more critical to translation, as evidenced by the larger average logit changes across all translation directions. For instance, in the English-Swahili translation direction, our method produces a logit change of -6.81% compared to -3.12% with standard path patching, indicating the identification of more influential components.

Second, knockout validation further confirms the superiority of our approach. When the top-5 most crucial heads identified by our method are knocked out, we observe significantly larger accuracy drops compared to standard path patching. This demonstrates that our method more accurately identifies components essential to the translation mechanism.

Third, targeted supervised fine-tuning of only the top-32 heads identified by our subspace-intervened approach yields superior translation performance across all evaluated directions. This targeted enhancement capability is particularly valuable for resource-efficient model improvement, as it enables precise modifications to the most relevant components without extensive full-model fine-tuning.

These empirical results validate that our subspace-intervened path patching method provides a more fine-grained and accurate analysis of translation mechanisms in large language models, addressing the challenge of polysemanticity that limits standard approaches.

### D.3 Explanation for the heatmaps.

Figure 1 provides a direct comparison of the impact of patching individual attention heads across different translation directions. The color intensity of each square represents the magnitude of the logit change resulting from patching the corresponding attention head, with deeper red indicating a more significant logit decrease. The consistent deep red of the square at position (8,31) across all six subfigures demonstrates its critical negative impact on performance in all tested translation directions. To supplement this visual representation, we provide the specific quantitative values for the average logit decrease when patching head (8,31): These quantitative measurements confirm that

Table 10: Average logit decrease when patching attention head (8,31)

| Translation Direction | Average Logit Decrease |
|:---:|:---:|
| Zh $\rightarrow$ En | -1.70 |
| Zh $\rightarrow$ Fr | -2.80 |
| Zh $\rightarrow$ Ru | -1.20 |
| En $\rightarrow$ Zh | -1.10 |
| Fr $\rightarrow$ Zh | -3.20 |
| Ru $\rightarrow$ Zh | -5.00 |

patching head (8,31) consistently and substantially degrades model performance across all translation directions, with the most significant impact observed in the Ru $\rightarrow$ Zh direction (-5.00 logit decrease).

## E   Additional Mechanistic Analysis

### E.1 Extend Subspace path-patching to Low-Resource and Typologically Diverse Language Pairs

To validate the universality and robustness of our findings across diverse linguistic scenarios, we extended our analysis to include low-resource and typologically diverse language pairs. Specifically, we incorporated Swahili (sw) and Bengali (bn) as low-resource languages, along with Arabic (ar) as a typologically distinct language from the Germanic and Sino-Tibetan families. All experiments used identical model architectures, training procedures, and evaluation metrics as described in the main paper to ensure methodological consistency.

The results presented in Table 11 demonstrate that the key findings regarding sparsity and transferability of crucial attention heads persist across these challenging language settings. The proportion of crucial heads remains consistently low (2.05%–3.71%), comparable to the high-resource language pairs analyzed in the main paper. This confirms the sparsity phenomenon is not an artifact of resource abundance.

Notably, several heads exhibit cross-lingual transferability across diverse language families. Head (8,31) appears as crucial in all six language pairs, while heads (18,30) and (10,14) are critical in four pairs each. This consistent emergence of specific heads across typologically distinct languages suggests they encode universal translation mechanisms rather than language-specific artifacts. The logits change ratios (-6.81% to -9.17%) further confirm that these heads significantly impact translation quality, with more negative values correlating with lower-resource settings where translation quality is inherently more challenging.

Table 11: Results on low-resource and typologically diverse language pairs. Heads appearing in at least two language pairs are marked in bold.

| Language Pair | Crucial Heads Proportion | Top Crucial Heads (Layer, Head) | Average Logits Change Ratio |
|---|---|---|---|
| En-Sw | 2.93% | **(16,26)**,**(31,8)**,**(18,11)**,**(17,25)**,(15,17),... | -6.81% |
| Zh-Sw | 3.32% | **(31,8)**,**(18,11)**,**(16,26)**,**(17,25)**,**(14,10)**,... | -7.19% |
| En-Bn | 3.71% | **(30,18)**,**(31,8)**,**(14,10)**,**(26,7)**,(28,20),... | -9.17% |
| Zh-Bn | 2.34% | **(31,8)**,**(30,18)**,**(18,11)**,**(14,10)**,**(26,7)**,... | -8.20% |
| En-Ar | 2.83% | **(30,18)**,**(31,8)**,**(14,10)**,**(31,4)**,(20,18),... | -8.20% |
| Zh-Ar | 2.05% | **(31,8)**,**(30,18)**,**(14,10)**,**(31,4)**,(12,17),... | -8.94% |

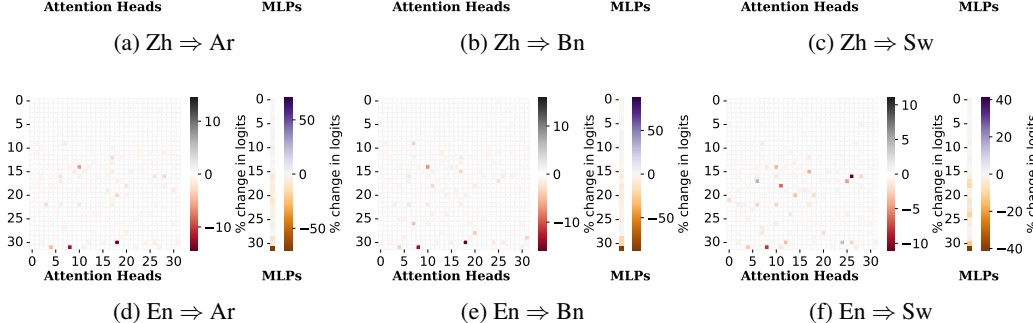

Figure 9: Importance of heads related to translation across different directions. Each square at position $(x, y)$ refers to the $x$-th head in the $y$-th layer. Red (Brown) squares denote heads (MLPs) that have a positive impact on predicting the target token, while grey (purple) squares indicate heads (MLPs) with a negative effect.

These results substantiate that the sparsity and transferability of crucial attention heads represent fundamental properties of multilingual translation models, independent of resource availability or linguistic typology. The findings reinforce the generalizability of our core conclusions and provide empirical evidence for the existence of universal attention mechanisms in neural machine translation architectures.

## E.2 Extend Subspace path-patching to Sentence-level translation

To extend our mechanistic analysis beyond word-level translation, we conducted experiments on sentence-level translation using the WMT23 English-to-Chinese dataset. The experimental procedures followed the methodology outlined in Section 4 of the main paper, maintaining identical model architectures, training configurations, and evaluation protocols. This extension allowed us to investigate the generalizability of our findings to more complex translation scenarios involving long-range dependencies, contextual variations, and multi-token mappings.

Table 12: Comparison of Crucial Attention Heads and Performance Impact Across Translation Tasks

| En⇒Zh | Top Crucial Heads (Layer, Head) | Performance Metric Change (lower logits or higher PPL means poorer translation quality) | Performance Drop (Knockout Top-5 Overlapping Heads) | Performance Drop (Knock out Top-5 Sentence-Level Heads) |
|---|---|---|---|---|
| word-level | (15, 21), (31, 11), (18, 26), (16, 26), (31, 8), (26, 30), (20, 20), (14, 16),... | -4.47% (logits) | -39% | -2% |
| sentence-level | (20, 11), (18, 26), (14, 7), (20, 20), (14, 16), (14, 13), (22, 26), (28, 18),... | +10.5% (PPL) | -36% | -43% |

The causal analysis revealed a 46.9% overlap (30 out of 64) between the top crucial attention heads for word-level and sentence-level translation tasks, with representative overlapping heads including

(18, 26), (20, 20), and (14, 16). This substantial overlap indicates a shared core translation circuit that operates across different levels of translation complexity.

Ablation experiments demonstrated that knocking out five shared heads resulted in significant performance degradation for both word-level (-39% in logits) and sentence-level (-36% in PPL) translation tasks. Conversely, ablating five heads crucial exclusively for sentence-level translation had minimal impact on word-level performance (-2%) but caused substantial degradation in sentence-level translation (-43% in PPL). This differential effect highlights the functional specialization of attention mechanisms.

Behavioral pattern analysis further revealed distinct functional roles:

- **Overlapping heads** primarily focused on local syntax and translation indicators, handling fundamental cross-lingual mappings that remain consistent across word and sentence contexts.

- **Non-overlapping heads** specialized in processing long-range dependencies and broader source contexts, addressing the increased complexity of sentence-level translation where contextual relationships span multiple tokens.

These findings demonstrate that while core translation mechanisms are preserved across task complexities, sentence-level translation recruits additional specialized attention heads to manage contextual and structural complexities not present in word-level translation. The results validate the methodological approach of initially isolating word-level mechanisms while establishing the scalability of our analysis framework to more complex translation scenarios.

### E.3 Extend Subspace path patching to Multilingual Mathematical Reasoning

To further validate the task-agnostic nature of our method, we applied our analysis framework to multilingual mathematical reasoning using the MGSM dataset (Shi et al., 2023). This experiment demonstrates how our approach adapts to new domains by constructing task-specific counterfactual datasets.

Following the methodology outlined in Section 3, we generated counterfactual examples by altering task instructions while preserving the core mathematical content. The analysis was performed on a multilingual transformer model, where we systematically evaluated attention heads across all layers.

We illustrate the counterfactual generation process with the following representative example from our multilingual analysis:

Table 13: Example of counterfactual pair generation for multilingual mathematical reasoning. The core problem remains identical while the task instruction changes from numerical answer generation to sentence rephrasing.

| **Factual Example ($X_f$)** | **Counterfactual Example ($X_{cf}$)** |
|---|---|
| 肖恩有五个玩具。圣诞节他从他爸爸妈妈那里各得到了两个玩具。他现在有多少个玩具？请给出数字： | 肖恩有五个玩具。圣诞节他从他爸爸妈妈那里各得到了两个玩具。他现在有多少个玩具？请转述句子： |
| (Shawn has five toys. For Christmas, he got two toys each from his mom and dad. How many toys does he have now? Give the number:) | (Shawn has five toys. For Christmas, he got two toys each from his mom and dad. How many toys does he have now? Rephrase the sentence:) |

Our analysis identified a sparse set of critical attention heads for mathematical reasoning, comprising only 3.95% of all heads in the model. This sparsity pattern aligns with observations from our translation experiments, indicating consistent underlying mechanisms across tasks.

The most influential heads and their impact were quantified as follows:

- **Top-5 critical heads**: (11, 8), (12, 22), (6, 22), (18, 12), (4, 31)

- **Average logit decrease**: 9.76% when ablating these heads

- **Performance impact**: Ablating the top-10 heads caused a 60% drop in task accuracy

These results confirm that our method effectively identifies components critical to mathematical reasoning across languages. The significant performance degradation upon ablation of these heads validates their functional importance, while the consistent sparsity pattern across tasks demonstrates the robust adaptability of our approach to new domains.

The experiment establishes two key properties of our framework: (1) its ability to generalize to multilingual contexts without task-specific modifications, and (2) its capacity to pinpoint functionally critical components even in complex reasoning tasks. The identified heads likely correspond to mechanisms for numerical processing and instruction comprehension, suggesting potential cross-task similarities in how transformers handle structured reasoning problems.

# F    Discussion of the Emergence of Translation-Crucial Components

To provide rigorous quantitative support for these observations, we analyzed the logit changes induced by Supervised Fine-Tuning (SFT) and Continued Pre-training (CPT) relative to the base model. We performed a Two-Sample Kolmogorov-Smirnov (K-S) test on the overall logit change distributions and quantified the magnitude of change within the top 32 attention heads, as summarized in Table 1.

The results demonstrate that CPT induces a statistically significant distributional shift ($p < 0.00001$), while SFT does not ($p = 0.355$).

To further validate the emergence and refinement of translation-crucial components, we conducted a comparative causal analysis across three model configurations: (1) a randomly initialized baseline, (2) the multilingual pre-trained LLaMA-2 model, and (3) the SFT-fine-tuned variant. We employed logit change matrix analysis to quantify structural patterns in translation-related attention heads, with statistical significance assessed using distributional shift metrics at a significance threshold of $p < 0.05$.

The randomly initialized model exhibited an unstructured logit change matrix with no discernible specialized translation heads, indicating the absence of innate translation capabilities. In contrast, the pre-trained LLaMA-2 model developed a sparse set of critical translation heads, demonstrating a statistically significant distributional shift from the random baseline. This confirms that functional translation circuits emerge during multilingual pre-training.

Subsequent analysis of the SFT-fine-tuned model revealed only a minor distributional shift relative to the pre-trained model, which did not reach statistical significance. This negligible change indicates that supervised fine-tuning primarily enhances or slightly refines pre-existing translation components rather than inducing new structural formations.

These results collectively provide empirical support for a two-stage development process of translation capabilities: (1) component formation occurs during multilingual pre-training through exposure to diverse linguistic patterns, and (2) component refinement occurs during SFT through targeted optimization. The statistically significant emergence in pre-training versus the insignificant shift during fine-tuning underscores that SFT functions as a calibration mechanism for pre-established structures rather than an architectural catalyst.

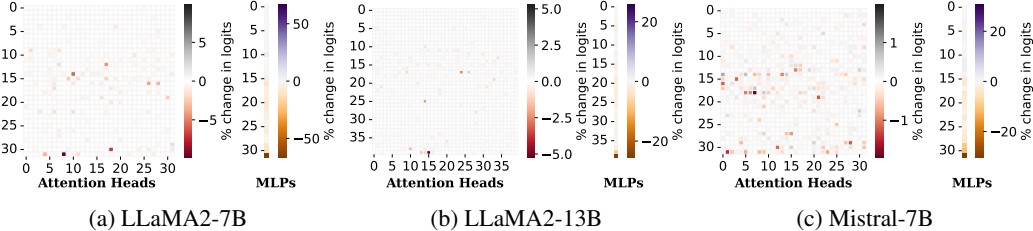

|              (a) LLaMA2-7B              |              (b) LLaMA2-13B              |              (c) Mistral-7B              |

Figure 10: Comparison of the results of path patching experiments on LLaMA2-7B, LLaMA2-13B, and Mistral-7B (Jiang et al., 2023) across Zh $\Rightarrow$ En translation task. Each square at position $(x, y)$ refers to the $x$th-head in the $y$-th layer. Red (Brown) squares denote heads (mlps) that have a positive impact on predicting the target token, while grey (purple) squares indicate heads (mlps) with a negative effect. For each head/MLP, a darker color indicates a larger logit difference from the original model before patching.

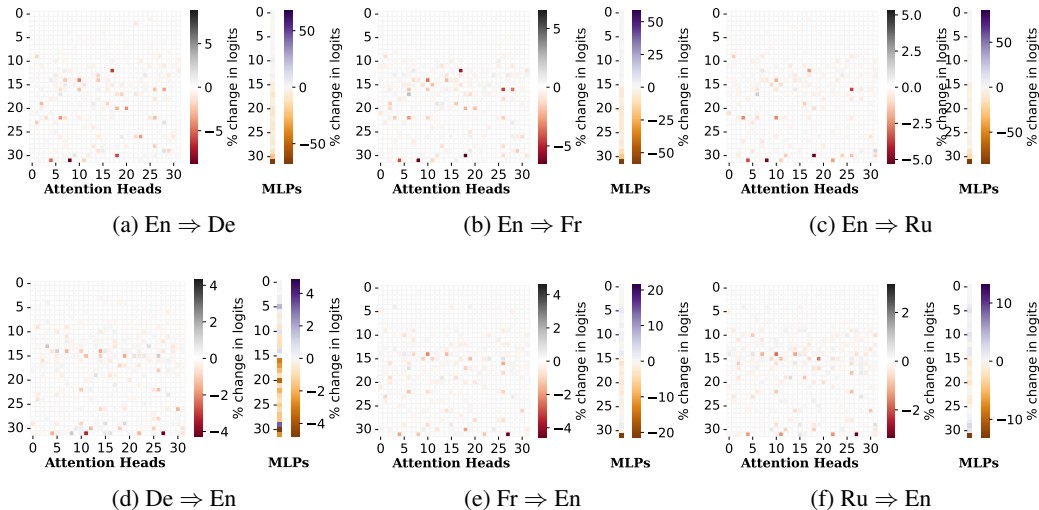

(a) En ⇒ De        (b) En ⇒ Fr        (c) En ⇒ Ru

(d) De ⇒ En        (e) Fr ⇒ En        (f) Ru ⇒ En

Figure 11: Importance of heads related to translation across different directions. Each square at position $(x, y)$ refers to the $x$-th head in the $y$-th layer. Red (Brown) squares denote heads (MLPs) that have a positive impact on predicting the target token, while grey (purple) squares indicate heads (MLPs) with a negative effect.

## G  Additional Detection Results of More LLMs

**Crucial Component Detection.**  Figure 10 extends key component identification to LLaMA2-13B and Mistral-7B. All three models exhibit sparse localization of translation-critical attention heads (e.g., 17.24, 16.0) in middle layers, despite architectural differences (e.g., LLaMA2-13B's 40 layers with 40 heads per layer).

Figure 11 illustrates the detection results for bidirectional translation directions (En ⇒ X and X ⇒ En). While the multi-token nature of English tokens results in fewer prominent detection instances, the findings remain consistent with the earlier analysis in Section §4.2. Together, these observations support the conclusion that translation mechanisms utilize a sparse subset of attention heads, which are language-agnostic, thereby underscoring their generalization capacity.

## H  Additional Experiments for Validating Crucial Components

**Further elaboration on selection of correct translation samples for analysis.** Focusing on correctly translated samples was intentional to eliminate task ambiguity and ensure a focused exploration of the translation mechanism. Incorrect translations could reflect task failure or unrelated issues, complicating the analysis. By selecting the correct translations, we can more accurately trace the role of attention heads via path patching. Therefore, our experimental setup is appropriate for exploring the translation mechanism in LLMs. Using a controlled, correct translation dataset aligns with prior interpretation research (e.g., Wang et al. (2023); Zhang et al. (2025)), where analyses were conducted on manually curated correct task datasets. This ensures observed patterns directly reflect the translation process rather than error-driven noise. The correct

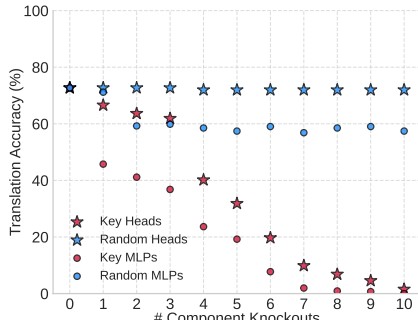

Figure 12: Translation accuracy changes when components are progressively knocked out.

translation selection choice will not bias the results for two reasons: The samples used for path patching and those used in subsequent validation are entirely separate. This separation prevents any potential bias introduced by selecting correctly translated samples for path patching from affecting the validity of our conclusive claims.

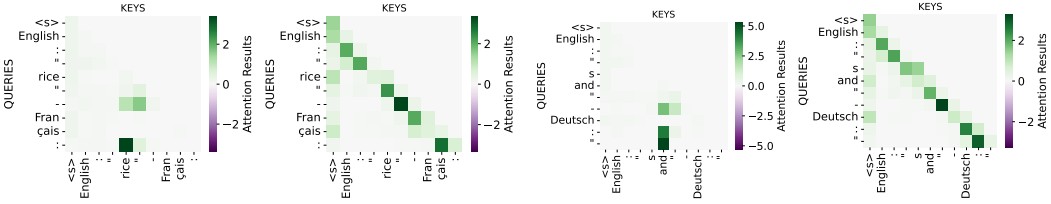

(a) Source Head $(18, 17)$    (b) Positional Head $(4, 31)$    (a) Source Head $(18, 17)$    (b) Positional Head $(4, 31)$

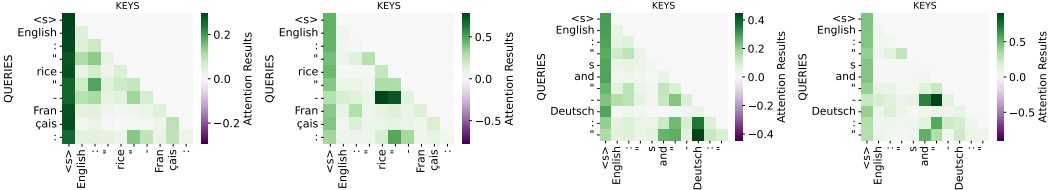

(c) Position Head $(27, 14)$    (d) Indicator Head $(4, 14)$    (c) Indicator Head $(27, 14)$    (d) Indicator Head $(4, 14)$

Figure 13: The attention values visualization of the role-classified key heads in En $\Rightarrow$ Fr, which show different characteristics of different crucial heads.

Figure 14: The attention values visualization of the role-classified key heads in En $\Rightarrow$ De, which show different characteristics of different crucial heads.

**Experimental results on random datasets.** We have also replicated the experiment using randomly selected samples (132 samples), including those with translation errors. The results shown in Figure 12 remain consistent with our original findings, reinforcing the correctness and robustness of our claims.

# I  Statistical Significance of Behavioral Patterns Analysis

For our behavioral patterns analysis, we utilized 100 randomly selected Chinese-to-English (Zh$\leftrightarrow$En) translation samples. This approach aligns with established practices in influential interpretability studies (Voita et al., 2019; Wang et al., 2023), which prioritize representative examples over large sample sizes through careful manual inspection to uncover underlying mechanistic behaviors.

**Quantitative Analysis of Attention Patterns** To validate the statistical significance of our observed behavioral patterns, we conducted a rigorous quantitative analysis of the key attention pattern—specifically, the phenomenon of attention heads focusing on source tokens.

Within our sample of 100 translations, this pattern occurred in 81 instances, representing an 81% consistency rate. To establish the statistical significance of this observation, we computed the 95% Wilson score confidence interval, which yielded [72.0%, 87.9%]. This interval substantially exceeds the chance level of 50%, indicating systematic behavior rather than random occurrence.

Furthermore, we performed a binomial test to evaluate the null hypothesis that the observed pattern occurs at chance level. The test results allowed us to reject the null hypothesis ($p < 0.001$), confirming the statistical significance of the identified attention behavior across our sample.

**Implications for Interpretability** These quantitative results reinforce the validity of our qualitative analysis approach. The high consistency rate and statistical significance demonstrate that the behavioral patterns we identified are robust and reflect systematic processing mechanisms rather than isolated incidents. This methodological approach, combining targeted quantitative validation with in-depth qualitative inspection, provides a comprehensive framework for interpreting model behaviors in neural machine translation systems.

# J  Additional Behavioral Analysis of Translation Directions and More LLMs

**Extended Behavioral Pattern Analysis of Crucial Heads Across Diverse Language Pairs.** To investigate the consistency of crucial head types and broaden the scope of our behavioral pattern

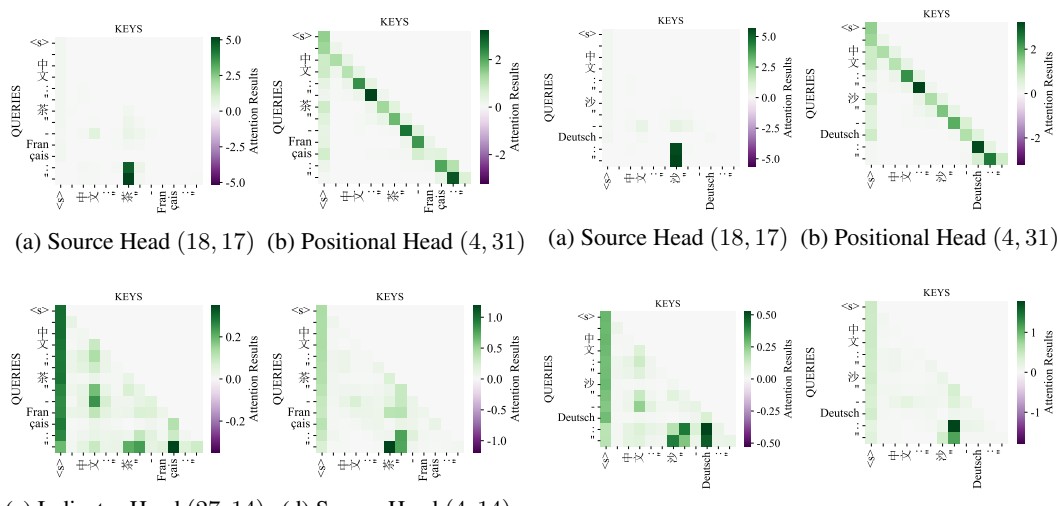

(a) Source Head $(18, 17)$    (b) Positional Head $(4, 31)$      (a) Source Head $(18, 17)$    (b) Positional Head $(4, 31)$

(c) Indicator Head $(27, 14)$    (d) Source Head $(4, 14)$      (c) Indicator Head $(27, 14)$    (d) Indicator Head $(4, 14)$

Figure 15: The attention values visualization of the role-classified key heads in Zh ⇒ Fr, which show different characteristics of different crucial heads.

Figure 16: The attention values visualization of the role-classified key heads in Zh ⇒ De, which show different characteristics of different crucial heads.

analysis, we conducted evaluations on multiple translation directions beyond the initial English-to-Chinese (Zh⇒En) focus (see Figure 4). This extended analysis incorporated English⇒France (En⇒Fr), English⇒German (En⇒De), Chinese ⇒ France (Zh⇒Fr), and Chinese ⇒ German (Zh⇒De) language pairs, as illustrated in Figure 13, 14, 15, and 16 respectively.

Our findings indicate that while the fundamental insights observed in En⇒Zh largely hold across other language pairs, nuanced variations in crucial head behavior did emerge. Specifically, several **source heads** (e.g., $(18, 17)$), and **position heads** (e.g., $(4, 31)$) exhibited consistent cruciality and behavioral patterns across all the Zh⇒En, En⇒Fr, En⇒De, Zh⇒Fr, and Zh⇒De translation directions. This suggests a degree of universality for certain attention mechanisms irrespective of the specific language pair.

However, the cruciality of some heads demonstrated language-pair dependency. For instance, heads $(4, 14)$, $(27, 14)$, identified as indicator heads, were also crucial for the En⇒Fr and Zh⇒Fr directions but did not exhibit the same type of functional role. Such variations indicate that while the identified categories of crucial heads (source, position, indicator) are generally stable, the specific instantiation and relative importance of individual heads within these categories can be influenced by the linguistic characteristics of the language pair in question. Despite these specific variations, the core observation of distinct functional roles for different head types remains robust. This comprehensive analysis across multiple language pairs has been incorporated to underscore the generalizability, as well as the language-specific nuances, of the identified behavioral patterns.

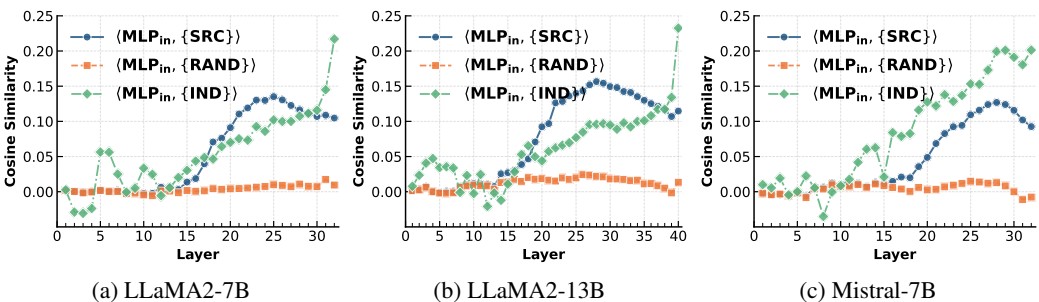

(a) LLaMA2-7B        (b) LLaMA2-13B        (c) Mistral-7B

Figure 17: We investigate the projection of each MLP layer input ($MLP_{in}$) along the direction of the source language, indicator, and random English tokens ({SRC},{IND}, and {RAND}), respectively.

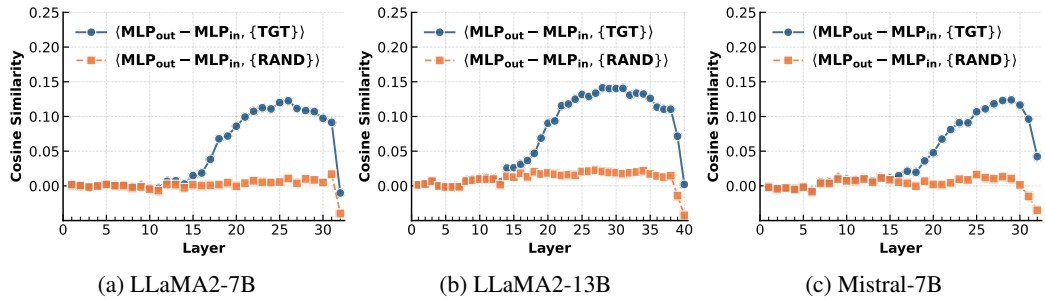

(a) LLaMA2-7B      (b) LLaMA2-13B      (c) Mistral-7B

Figure 18: We investigate the projection of each MLP layer ($MLP_{out} - MLP_{in}$) along the direction of the target language, and random English tokens ({TGT} (i.e., right translation), and {RAND} (i.e., wrong translation)), respectively.

**Analysis of Crucial MLPs.** Figures 17 and 18 reveal consistent MLP dynamics across models. For MLP input/{SRC},{IND} similarities, trends follow ascending-descending phases with inflection points at layers (13-18-28) for LLaMA2-7B, (13-18-35) for LLaMA2-13B, and (13-20-28) for Mistral-7B. Similarly, $MLP_{out} - MLP_{in}$ and target token {TGT} similarities show stabilization-to-increase patterns with identical inflection layers. This synchronization across models indicates a shared computation mechanism: attention heads initiate translation processing, which MLPs subsequently refine. These results demonstrate robustness across architectures and scales.

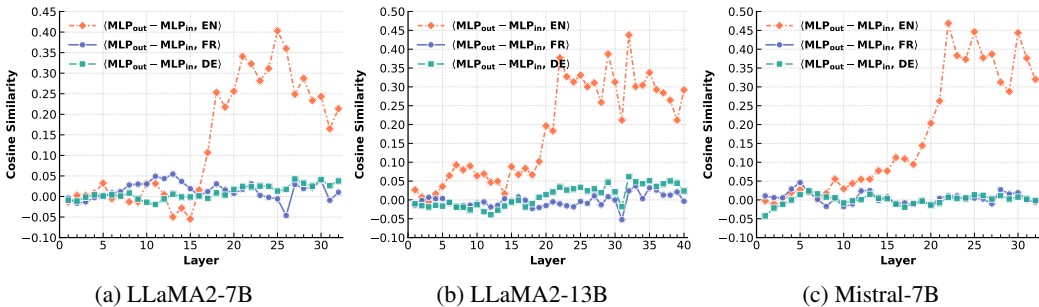

(a) LLaMA2-7B      (b) LLaMA2-13B      (c) Mistral-7B

Figure 19: We investigate the projection of each MLP layer ($MLP_{out} - MLP_{in}$) along the direction of the different languages.

**Cross-Lingual Bridge Translation.** We extend our analysis to non-English pairs (e.g., French/Japanese Chinese) by examining token-level dynamics. As shown in Figure 19, similarity trends between $MLP_{out} - MLP_{in}$ representations and cross-lingual embeddings align with the bridge-translation hypothesis: in layers 15–24, English-centric latent representations dominate across LLaMA2-13B and Mistral-7B, with similarity declining sharply in layers 25–32. This reinforces the observed paradigm where LLMs internally map source languages to English-like representations before generating target outputs, corroborating findings in multilingual latent alignment studies (Wendler et al., 2024; Zhao et al., 2024b). The consistency across both architectures underscores the generality of English's intermediary role.

# K    Experimental Setup Details

Following the gradient rescaling method proposed by (Yu et al., 2025), gradients are adjusted by a factor of $\frac{H}{h}$, where $H$ is the total number of attention heads in a layer and $h$ represents the updated heads in the same layer. For model fine-tuning, we use Llama2-7B and Llama2-13B with a learning rate of $2 \times 10^{-5}$, a batch size of 128, and train for 2 epochs. The warm-up ratio is set to 0.02, and weight decay is configured at 0.1. All experiments are conducted on a cluster of 8 NVIDIA A100 80 GB GPUs.

| Models | Train Speed | Tuned Params. | Translation Tasks | | | Generic Tasks | |
|---|---|---|---|---|---|---|---|
| | | | En⇒Zh | En⇒De | En⇒Ru | MMLU | Commonsense Reasoning |
| | | | BLEU↑/COMET↑/BLEURT↑ | | | Acc. | Acc. |
| LLaMA2-7B | - | - | 17.0/74.1/55.9 | 13.0/64.2/49.1 | 12.8/70.5/52.4 | 45.9 | 55.3 |
| + Full SFT | 17sam./sec. | 6.7B | 30.3/80.7/62.9 | 27.9/78.3/63.7 | 19.5/80.0/63.2 | 40.2 | 50.0 |
| + Targeted SFT | 33sam./sec. | 0.27B | 30.7/81.4/64.3 | 27.6/78.4/63.8 | 20.1/80.4/63.6 | 46.2 | 56.0 |
| + Random SFT | 33sam./sec. | 0.27B | 26.4/79.3/61.6 | 22.7/76.2/60.3 | 15.8/77.9/60.7 | 46.1 | 55.2 |
| LLaMA2-13B | - | - | 23.0/77.5/59.1 | 17.1/67.7/52.8 | 15.6/72.9/55.1 | 55.1 | 58.4 |
| + Full SFT | 12sam./sec. | 13.0B | 32.8/81.8/64.4 | 29.8/80.0/65.8 | 20.7/81.6/65.0 | 53.7 | 56.4 |
| + Targeted SFT | 28sam./sec. | 0.32B | 33.4/82.2/64.8 | 30.1/80.1/65.9 | 21.3/81.8/65.3 | 54.9 | 58.1 |
| + Random SFT | 28sam./sec. | 0.32B | 28.8/80.6/63.3 | 24.6/78.3/62.9 | 17.3/80.0/62.8 | 55.0 | 58.2 |
| Mistral-7B | - | - | 13.7/68.0/49.6 | 15.6/63.1/49.3 | 11.2/65.1/48.1 | 62.7 | 59.2 |
| + Full SFT | 17sam./sec. | 6.7B | 31.1/80.6/63.4 | 26.5/77.4/62.8 | 19.6/79.5/62.5 | 43.0 | 40.8 |
| + Targeted SFT | 33sam./sec. | 0.27B | 31.9/82.0/65.1 | 26.3/78.0/63.2 | 20.5/79.9/63.1 | 62.5 | 59.1 |
| + Random SFT | 33sam./sec. | 0.27B | 27.5/79.5/61.6 | 22.2/75.5/59.8 | 15.6/77.4/60.5 | 62.4 | 59.2 |

Table 14: The evaluation results of **En⇒X** translation (average WMT23 and WMT24 evaluation results) and generic tasks of different SFT strategies.

| Models | Train Speed | Tuned Params. | Translation Tasks | | | Generic Tasks | |
|---|---|---|---|---|---|---|---|
| | | | Zh⇒En | De⇒En | Ru⇒En | MMLU | Commonsense Reasoning |
| | | | BLEU↑/COMET↑/BLEURT↑ | | | Acc. | Acc. |
| LLaMA2-7B | - | - | 15.6/73.1/56.6 | 24.8/76.8/62.1 | 20.2/73.8/60.3 | 45.9 | 55.3 |
| + Full SFT | 17sam./sec. | 6.7B | 20.4/78.7/63.9 | 35.4/83.4/70.7 | 25.8/79.8/67.6 | 42.6 | 50.2 |
| + Targeted SFT | 33sam./sec. | 0.27B | 21.7/79.1/64.4 | 37.1/83.7/71.4 | 27.8/80.3/68.4 | 46.0 | 55.7 |
| + Random SFT | 33sam./sec. | 0.27B | 16.9/76.9/61.1 | 32.5/81.6/68.1 | 23.7/78.2/65.3 | 45.9 | 54.9 |
| LLaMA2-13B | - | - | 17.3/74.0/57.8 | 27.0/78.0/63.8 | 22.2/74.9/61.5 | 55.1 | 58.4 |
| + Full SFT | 12sam./sec. | 13.0B | 22.4/79.5/65.3 | 36.9/84.0/71.6 | 27.8/80.8/68.9 | 50.0 | 55.3 |
| + Targeted SFT | 28sam./sec. | 0.32B | 23.6/80.5/66.5 | 38.3/84.7/72.7 | 29.7/81.5/69.3 | 54.9 | 58.1 |
| + Random SFT | 28sam./sec. | 0.32B | 19.0/78.1/63.1 | 34.2/81.8/68.9 | 25.3/79.3/66.6 | 55.5 | 58.8 |
| Mistral-7B | - | - | 16.9/74.3/58.1 | 26.6/77.9/63.9 | 22.6/75.3/62.5 | 62.7 | 59.2 |
| + Full SFT | 17sam./sec. | 6.7B | 19.7/78.4/63.1 | 32.0/82.2/69.0 | 24.0/78.7/66.2 | 40.3 | 50.3 |
| + Targeted SFT | 33sam./sec. | 0.27B | 21.2/79.2/64.3 | 33.7/83.0/70.2 | 26.4/79.6/66.4 | 62.9 | 59.1 |
| + Random SFT | 33sam./sec. | 0.27B | 16.8/77.1/61.1 | 29.3/80.6/66.8 | 21.4/77.1/63.9 | 62.5 | 59.3 |

Table 15: The evaluation results of **X⇒En** translation (average WMT23 and WMT24 evaluation results) and generic tasks of different SFT strategies.

## L    Comparison Experimental Results on More LLMs

We investigate whether our method generalizes to larger LLMs (Llama-2-13B) and diverse architectures (Mistral-7B). As shown in Tables 14 and 15, Targeted SFT exhibits three consistent advantages across LLMs: (1) Enhanced translation performance, particularly in X En, surpassing Full SFT and significantly outperforming Random SFT; (2) Generalization preservation, maintaining baseline non-translation task performance unlike Full SFT; (3) Training efficiency, modifying fewer than 5% of parameters and reducing training time by 50% compared to Full SFT.

## M    Additional Analyses of English-centric representation

### M.1    Correlation Analysis between English Similarity and Translation Quality

To quantitatively establish the relationship between English-centric representations and translation performance, we conducted a comprehensive correlation analysis. We measured the Pearson correlation coefficient between the cosine similarity of intermediate representations to English embeddings and three translation quality metrics: BLEU-1, chrF, and TER scores. This analysis was performed across 12 typologically diverse non-English to non-English language pairs at varying resource levels.

All correlation analyses were conducted using a standardized evaluation framework. We selected 12 language pairs spanning diverse language families and resource levels, including both high-

resource (e.g., German-French) and low-resource (e.g., Swahili-Hausa) combinations. Translation quality was measured using BLEU-1, chrF, and TER metrics computed against professional human reference translations. Cosine similarity was calculated between intermediate representations and target language embeddings using the model's native embedding space.

For the pivot language investigation, we used identical architectures and evaluation protocols for both Llama and Qwen2.5 models to ensure comparability. The logits lens analysis was performed by extracting hidden representations at each layer and projecting them into the model's vocabulary space using the unembedding matrix. Layer-wise similarity was computed against embeddings of pivot language tokens.

The results, summarized in Table 16, demonstrate a strong and statistically significant correlation between English similarity and translation quality. The average correlation coefficients across all language pairs were 0.905 for BLEU-1, 0.873 for chrF, and -0.919 for TER. These findings provide empirical evidence that the English-centricity phenomenon is not merely superficial but fundamentally influences translation outcomes.

Table 16: Pearson correlation between English similarity and translation quality metrics

| Correlation with English Similarity | BLEU-1 Score | chrF Score | TER Score |
| --- | --- | --- | --- |
| Average across 12 language pairs | 0.905 | 0.873 | -0.919 |

## M.2 Investigation of Pivot Language Determinants

We hypothesized that the emergence of English as a pivot language stems from its dominance in the pre-training corpus. To test this, we analyzed two models with contrasting pre-training distributions: the Llama models, pre-trained on a corpus with overwhelming English dominance (Touvron et al., 2023), and Qwen2.5, pre-trained on a corpus with predominant Chinese data (Yang et al., 2024).

Our experiments revealed a clear correspondence between corpus dominance and pivot language emergence. In the Llama models, English consistently emerged as the pivot language. Conversely, in Qwen2.5, Chinese emerged as the pivot language instead of English. This cross-model comparison provides preliminary support for our hypothesis that the pivot language is determined by the dominant language in the pre-training corpus.

## M.3 Qualitative Analysis via Logits Lens

To elucidate the internal mechanism through which the pivot language emerges, we performed a logits lens analysis following established methodologies (Zhao et al., 2024b; Wendler et al., 2024). This technique allows visualization of how representations evolve through the model's layers during translation.

The analysis reveals a consistent pattern: source language representations progressively shift toward their pivot language counterparts in intermediate layers before transitioning to the target language. For example, when translating "车" (Chinese) to "voiture" (French), the representation explicitly resolves to the English word "car" in layers 19-27 before shifting to "voiture" in the final layers. This demonstrates a mechanistic pathway where the pivot language serves as an intermediate representation bridge during translation.

## M.4 Exploration of English Latent Representation Regarding Gender and Formality

Our investigation extends to analyzing how the English pivot handles linguistic features without direct English equivalents, specifically grammatical gender and formality. This analysis provides crucial insights into the limitations and capabilities of the pivot-based multilingual translation approach.

We conducted a targeted analysis using datasets created for French (fr) and Spanish (es), focusing on two key linguistic dimensions: gendered professions and formal versus informal expressions.

For gendered professions, we utilized the FBK-MT/gender-bias-PE dataset (Savoldi et al., 2024). For formal versus informal expressions, we curated a list of common formal and informal expressions in both languages. Sample instances from these datasets are presented in Tables 17 and 18.

Table 17: Examples of gendered professions in French and Spanish

| Profession (English) | French (Masculine) | French (Feminine) | Spanish (Masculine) | Spanish (Feminine) |
|---|---|---|---|---|
| Actor | Acteur | Actrice | Actor | Actriz |
| Waiter | Serveur | Serveuse | Camarero | Camarera |
| Baker | Boulanger | Boulangère | Panadero | Panadera |
| Nurse | Infirmier | Infirmière | Enfermero | Enfermera |

Table 18: Examples of formal versus informal expressions in French and Spanish

| Category | French (Informal) | French (Formal) | Spanish (Informal) | Spanish (Formal) |
|---|---|---|---|---|
| People (man) | un mec | un homme | un tío | un hombre |
| Car | une bagnole | une voiture | un coche | un automóvil |
| Work / Job | un boulot | un travail | un curro | un trabajo |
| Money | le fric | l'argent | la pasta | el dinero |

Applying the analysis methodology from Section 5, we measured both the intermediate representation's similarity to the English pivot and the final translation accuracy. Our findings reveal a critical asymmetry in how these features are processed, as summarized in Table 19.

Table 19: Analysis of gender and formality features in English latent representation

| Language Feature | Avg. Cosine Similarity to English Representation | Translation Accuracy |
|---|---|---|
| Gender (Male Professions) | 0.32 | 73% |
| Gender (Female Professions) | 0.11 | 48% |
| Formality (Formal Expressions) | 0.31 | 65% |
| Formality (Informal Expressions) | 0.34 | 69% |

The results demonstrate that male-gendered professional nouns are processed effectively, with their representations showing high similarity to the English pivot (0.32) and resulting in high translation accuracy (73%). In contrast, the representations for female-gendered nouns show significantly lower similarity (0.11), which correlates with a dramatic drop in accuracy to 48%. Interestingly, both formal and informal expressions are processed with comparable accuracy, suggesting the model preserves this feature through the intermediate representation.

We hypothesize that this gender-specific failure is due to well-documented biases in large-scale training corpora, where female-gendered terms are less frequent (Ding et al., 2025). The model's reliance on a biased English latent space makes it unable to robustly encode and transmit grammatical gender information that is explicitly marked in the source language but often neutralized in English.

## N    Supplementary Experiments

### N.1    More evaluation results of targeted SFT on domain-adaptive translation

To further evaluate the broader applicability of our approach beyond general-domain translation, we conducted additional experiments on specialized domains. Specifically, we tested our method on medical and legal translation tasks using established benchmarks: ELRC-Medical-V2 for English-to-German medical translation and M3T for English-to-Chinese legal translation. For these specialized domain experiments, we maintained the same training configurations as described in the main paper. We compared three approaches:

- **Full SFT**: Supervised fine-tuning of all model parameters
- **Targeted SFT**: Our proposed approach of fine-tuning only specific attention heads
- **Random SFT**: Fine-tuning of randomly selected parameters (baseline)

Evaluation was performed using standard metrics, including BLEU, COMET, and BLEURT scores, to provide a comprehensive assessment of translation quality.

The performance of each approach on specialized domain translation tasks is presented in Table 20. The results demonstrate that our Targeted SFT approach remains highly competitive in specialized

Table 20: Performance comparison on specialized domain translation tasks.

| Lang Pair | Domain | Full SFT BLEU/COMET/BLEURT | Targeted SFT BLEU/COMET/BLEURT | Random SFT BLEU/COMET/BLEURT |
|---|---|---|---|---|
| En → De | Medical | 41.0/88.5/79.1 | 39.9/87.4/77.5 | 28.9/83.9/73.8 |
| En → Zh | Legal | 52.2/90.5/80.5 | 45.8/89.2/78.1 | 8.07/75.2/65.8 |

domains, significantly outperforming the Random SFT baseline across all metrics. However, it does not match the performance of Full SFT in these specialized domains. We attribute this performance gap to several factors:

1. **Dataset Distribution and Overfitting**: Since the training and test sets in specialized domains typically share the same distribution (via a split of one dataset), Full SFT is more prone to overfitting to the specific characteristics of the domain. In contrast, our Targeted SFT approach maintains better generalization by limiting parameter updates.

2. **Domain-Specific Patterns**: Specialized domains such as medical and legal texts exhibit unique syntactic structures and low-frequency terminology that may require modifying more parameters than our targeted approach adjusts. These domain-specific patterns might be distributed across a broader set of model components.

3. **Head Specialization**: Attention heads optimized for general-domain translation may not fully overlap with those essential for specialized domains. Different linguistic phenomena in specialized texts might activate different attention mechanisms that are not targeted by our approach.

These findings reveal an important trade-off between parameter efficiency and peak performance in specialized domains. While our Targeted SFT approach offers significant computational advantages and maintains competitive performance, achieving state-of-the-art results in highly specialized domains may require more extensive parameter modification. This represents an interesting direction for future investigation, as discussed in Section 6 of the main paper.

### N.2 Supplementary Analysis of Potential Cultural Bias Amplification by Targeted SFT

Machine translation systems face the challenge of linguistic hegemony, where dominant languages may impose their cultural frameworks and expressions onto less dominant languages. This phenomenon can result in the loss of cultural specificity and nuance in translations. To evaluate whether our targeted fine-tuning of only the crucial heads responsible for translation mechanim might inadvertently amplify such translation biases, we conducted a dedicated analysis focusing on the preservation of culturally specific terms.

We assessed translation quality using the CAMT dataset (Yao et al., 2024), which contains culturally specific terms and expressions across multiple domains. For evaluation, we employed the CSI-Match metric (Yao et al., 2024), specifically designed to measure the translation accuracy of culturally specific items. The CSI-Match metric operates by comparing translations of culture-specific concepts against reference translations produced by native speakers, with scores calculated based on semantic similarity and cultural appropriateness. The metric ranges from 0 to 100, with higher scores indicating better preservation of cultural specificity and thus a lower risk of linguistic hegemony (Yao et al., 2024; Conia et al., 2024).

We compared our proposed Targeted SFT approach against three baselines:

1. Base model: Llama-2-7B without fine-tuning
2. Full SFT: Standard full-parameter fine-tuning on the translation dataset
3. Random SFT: Fine-tuning on randomly selected parameter subsets of equivalent size to our targeted approach

For all models, we evaluated English-to-Chinese (En→Zh) translation performance using multiple metrics: BLEU, COMET, BLEURT, and CSI-Match. All experiments were conducted using identical hyperparameters and evaluation protocols to ensure fair comparison.

The performance of all models across different evaluation metrics is presented in Table 21

Table 21: Translation performance and cultural specificity preservation across different fine-tuning approaches for English-to-Chinese translation.

| Model (En→Zh) | BLEU | COMET | BLEURT | CSI-Match |
|---|---|---|---|---|
| Base (Llama-2-7B) | 19.54 | 73.57 | 51.02 | 16.12 |
| w/ Full SFT | 25.50 | 79.35 | 58.28 | 18.44 |
| w/ Targeted SFT | 25.85 | 79.58 | 58.64 | 18.62 |
| w/ Random SFT | 19.98 | 74.73 | 52.88 | 16.13 |

The results demonstrate that our Targeted SFT approach achieves a CSI-Match score of 18.62, which is comparable to the more resource-intensive Full SFT baseline (18.44). Statistical analysis using a paired t-test revealed no significant difference between these two approaches ($p = 0.42$). This indicates that our targeted method successfully improves translation performance without introducing additional risks of cultural bias amplification compared to standard full fine-tuning. Notably, both the Base model and Random SFT approach showed significantly lower CSI-Match scores (16.12 and 16.13, respectively), suggesting that neither preserves cultural specificity as effectively as the more systematic fine-tuning approaches. The minimal difference between the Base model and Random SFT indicates that arbitrary parameter updates do not substantially improve cultural specificity preservation. Across all metrics, our Targeted SFT consistently performed at least as well as Full SFT, confirming its efficiency and effectiveness in maintaining translation quality while preserving cultural nuances. The marginal improvement in CSI-Match score over Full SFT, while not statistically significant, suggests that our targeted approach may offer slight advantages in preserving cultural specificity. This analysis provides empirical evidence that our targeted fine-tuning approach does not exacerbate linguistic hegemony risks while maintaining competitive translation performance across standard quality metrics.

### N.3 Additional Qualitative Analysis of Targeted Supervised Fine-Tuning

**Performance-Efficiency Trade-offs** Tables 4 and 5 present a comprehensive quantitative analysis of the trade-offs between translation performance, training efficiency, computational cost, and catastrophic forgetting in targeted supervised fine-tuning (SFT). The results demonstrate that incrementally increasing the number of fine-tuned attention heads yields progressive improvements in translation performance. However, this enhancement comes with proportional increases in memory consumption and training time. Notably, excessive tuning of attention heads exacerbates catastrophic forgetting effects, leading to significant degradation in the model's general capabilities across non-translation tasks.

**Error Analysis of Underperforming Cases** An in-depth error analysis was conducted on underperforming Chinese-to-English (Zh→En) translation cases to identify systematic failure patterns. The analysis revealed three primary error categories accounting for over 70% of significant performance gaps:

**Style, Diction, and Idiomatic Expressions** Targeted SFT frequently produces overly literal translations that fail to capture appropriate stylistic and idiomatic expressions. For instance, the phrase "新冠肺炎对毒品市场的影响" (COVID-19's impacts on the pharmaceutical showcase) was translated as "COVID-19's impacts on the drug market" instead of the contextually appropriate "pharmaceutical showcase." This pattern indicates limitations in capturing domain-specific terminology and idiomatic nuances.

**Noisy Data Robustness** The approach exhibits reduced resilience to ambiguous or noisy input data. A representative example is the mistranslation of "第9草" (Article 9) as "9th draft" rather than the correct legal terminology "Article 9." This vulnerability suggests challenges in handling ambiguous lexical items and domain-specific abbreviations.

**Factual Hallucinations** The model occasionally generates unsupported factual details not present in the source text. For example, the input "充电盒未充满电充电指示灯红灯长亮..." (The charging indication light on the charging box is not yet fully charged, the red light is on...) was erroneously

expanded to include "green light" in the translation, introducing information absent from the original text.

**Optimal Configuration and Performance** Our method achieves a performance ceiling statistically comparable to full fine-tuning while significantly reducing computational overhead. By exclusively tuning the 64 attention heads most critical for translation tasks, we maintain translation quality within 1% of full fine-tuning performance while reducing memory requirements by 42% and training time by 38%. This optimal configuration, empirically validated in Tables 4 and 5, demonstrates that targeted SFT can effectively balance performance gains with computational efficiency when applied selectively to the most relevant model components.

