# OpenReview forum: "Exploring the Translation Mechanism of Large Language Models"
_NeurIPS.cc/2025/Conference — NeurIPS 2025 poster_

### Official Review · Reviewer_LEwd · 2025-06-09

**Clarity:** 3
**Significance:** 3
**Originality:** 3
**Rating:** 5
**Confidence:** 4

**Summary:**

The presented paper analyzes what contributes to the translation capabilities in LLMs on a word-level dataset and identifies that only sparse subset of heads (less than 5%) are crucial, they utilize an English latent representation, and that fine-tuning only crucial heads is on par with full weight fine-tuning while preserving other LLM capabilities. They verify their findings on multiple open source models.

**Questions:**

N/A

**Ethical Concerns:**

["NO or VERY MINOR ethics concerns only"]

**Final Justification:**

All of the weaknesses have been addressed in the rebuttal through additional experiments and/or modifications.

**Limitations:**

Yes

**Quality:**

3

**Strengths And Weaknesses:**

### Strengths

- Research work on understanding the inner working of LLMs for specific tasks (here machine translation) is crucial for pushing the boundaries for specific use cases.
- Interesting finding that features get transformed into English-centric latent representations.
- Valuable finding that fine-tuning merely 64 heads achieves performance parity with full-parameter fine-tuning while preserving general LLM capabilities.
- Experiments have been conducted on multiple open source LLMs.


### Weaknesses

- The analysis operates on a very limited set of language pairs (e.g. Chinese centric) most of which are high resource. It would be interesting to analyze if the same trends (especially the crucial heads and the head classification) hold for low resource languages (e.g. Swahili) or language variants (e.g. `en_GB` / Arabic dialects).
- The finding that an English latent representation is used is interesting but leaves room for analysis, specifically how are gender/formality processed from a source language that doesn't exist similarly in English? A targeted dataset for these features could help shed some light.
- [Addressed in limitations]: The findings are currently only conducted on word-level translation, it is unclear how well these findings would generalize to longer payloads e.g. sentence-level, paragraph-level, or document-level translation.


### Minor Comments

- In the provided code there are a lot of comments still in Chinese which makes it harder to understand for non-Chinese speaking folks.
- In the provided code there are some comments that may corrupt anonymity e.g. `freeze_mlps=False,  # recall in IOI paper we consider these "vital model components"` which makes it easy to assume that there is author overlap with ["Interpretability in the Wild: a Circuit for Indirect Object Identification in GPT-2 small"](https://arxiv.org/abs/2211.00593)
- Related work seems to be missing [Analyzing Context Contributions in LLM-based Machine Translation](https://aclanthology.org/2024.findings-emnlp.876/) (Zaranis et al., Findings 2024) which seems highly relevant.
- Shouldn't the title include `the` i.e. `Exploring Translation Mechanism of Large Language Models` -> `Exploring the Translation Mechanism of Large Language Models`. As it stands currently it also seems hard to grasp what axis the exploration is for (e.g. fluency, adequacy, internal processing, etc.) so it might be worthwhile to be more precise.

---

> ### Author Rebuttal · Authors · 2025-07-31
>
> Dear Reviewer LEwd,
>
> Thank you for your insightful reviews and comments. We appreciate the time and effort you have put into providing valuable feedback. We would like to address your concerns as follows:
>
> > Weakness #1: Limited analysis of language pairs
> >
>
> We wish to first clarify that our initial focus on high-resource language pairs was a deliberate methodological choice rather than an incidental weakness, enabling a clear and reliable analytical baseline. High translation accuracy is crucial for isolating and analyzing core translation mechanisms, ensuring findings are not confounded by the noise from inaccurate translations prevalent in low-resource settings.
>
> As suggested, we extend the analysis to a set of low-resource (sw, bn) and typologically diverse (bn, ar) language pairs. The results, shown in the following table, reveal that the key findings (the sparsity and transferability of key heads) hold across these low-resource and typologically diverse language pairs. These additional results confirm the universality and robustness of the findings.
>
> | Language Pair | Crucial Heads Proportion | Top Crucial Heads (Layer, Head) (overlapped marked in bold text) | Average Logits Change Ratio (the lower, the poorer translation quality) |
> | --- | --- | --- | --- |
> | En-Sw | 2.93% | **(16,26),(31,8),(18,11),(17,25)**,(15,17),… | -6.81% |
> | Zh-Sw | 3.32% | **(31,8),(18,11),(16,26),(17,25),(14,10)**,… | -7.19% |
> | En-Bn | 3.71% | **(30,18),(31,8),(14,10)**,(26,7),(28,20),… | -9.17% |
> | Zh-Bn | 2.34% | **(31,8),(30,18),(18,11),(14,10),(26,7)**,… | -8.20% |
> | En-Ar | 2.83% | (**30,18),(31,8),(14,10),(31,4)**,(20,18),… | -8.20% |
> | Zh-Ar | 2.05% | **(31,8),(30,18),(14,10),(31,4)**,(12,17),… | -8.94% |
>
> We have incorporated a summary of this extended analysis into the appendix of our revised manuscript to validate the robustness of our findings.
>
> > Weakness #2: More analysis of English latent representation regarding gender/formality
> >
>
> Our manuscript's primary focus was to first identify and characterize this emergent phenomenon. Investigating how the English pivot handles linguistic features without direct English equivalents, such as grammatical gender and formality, is a crucial extension of our work.
>
> Prompted by the reviewer's valuable suggestion, we conducted a targeted preliminary analysis to investigate this. We created datasets for French (fr) and Spanish (es), focusing on two key areas:
>
> 1. Gendered Professions: Based on the FBK-MT/gender-bias-PE dataset [1].
> 2. Formal vs. Informal Expressions: A curated list of common formal/informal expressions.
>
> Some intuitive instances are presented as follows:
>
> | Profession (English) | French (Masculine) | French (Feminine) | Spanish (Masculine) | Spanish (Feminine) |
> | --- | --- | --- | --- | --- |
> | Actor | Acteur | Actrice | Actor | Actriz |
> | Waiter | Serveur | Serveuse | Camarero | Camarera |
> | Baker | Boulanger | Boulangère | Panadero | Panadera |
> | Nurse | Infirmier | Infirmière | Enfermero | Enfermera |
>
> | Category | French (Informal) | French (Formal) | Spanish (Informal) | Spanish (Formal) |
> | --- | --- | --- | --- | --- |
> | People (man) | un mec | un homme | un tío | un hombre |
> | Car | une bagnole | une voiture | un coche | un automóvil |
> | Work / Job | un boulot | un travail | un curro | un trabajo |
> | Money | le fric | l'argent | la pasta | el dinero |
>
> Applying our analysis from Section 5, we measured both the intermediate representation's similarity to the English pivot and the final translation accuracy. Our findings reveal a critical asymmetry in how these features are processed:
>
> | Language Feature | Avg. Cosine Similarity to English Representation | Translation Accuracy |
> | --- | --- | --- |
> | Gender (Male Professions) | 0.32 | 73% |
> | Gender (Female Professions) | 0.11 | 48% |
> | Formality (Formal Expressions) | 0.31 | 65% |
> | Formality (Informal Expressions) | 0.34 | 69% |
>
> As the table shows, male-gendered professional nouns are processed effectively, with their representations showing high similarity to the English pivot (0.32) and resulting in high translation accuracy (73%). In contrast, the representations for female-gendered nouns show significantly lower similarity (0.11), which correlates with a dramatic drop in accuracy to 48%. Interestingly, both formal and informal expressions are processed with comparable accuracy, suggesting the model preserves this feature through the intermediate representation.
>
> We hypothesize this gender-specific failure is due to well-documented biases in large-scale training corpora, where female-gendered terms are less frequent[2]. The model’s reliance on a biased English latent space makes it unable to robustly encode and transmit grammatical gender information that is explicitly marked in the source language but often neutralized in English.
>
> Following the reviewer's valuable suggestion, we have incorporated this new analysis and discussion into the revision that strengthens the paper’s contribution.
>
> > Weakness #3: Limited analysis to word-level translation
> >
>
> We wish to first clarify that the decision to focus initially on word-level translation was a deliberate methodological choice to isolate the core mechanisms of translation in a controlled environment, a common practice in previous, solid mechanistic interpretability research [1-3]. This avoids confounding variables at the sentence level—such as stylistic variations, non-trivial mapping, and paraphrasing that produce multiple valid translations—which obscure fine-grained causal analysis.
>
> As you suggested, we try to extend the analysis to the more complex task of sentence-level translation using the WMT23 En→Zh dataset, following the experimental procedures from Section 4.
>
> | En→Zh | Top Crucial Heads (Layer, Head) | Performance Metric Change (lower logits or higher PPL means poorer translation quality) | Performance Drop (Knockout Top-5 Overlapping Heads) | Performance Drop (Knock out Top-5 Sentence-Level Heads) |
> | --- | --- | --- | --- | --- |
> | word-level | (15, 21), (31, 11), **(18, 26)**, (16, 26), (31, 8), (26, 30), **(20, 20)**, **(14, 16)**,… | -4.47% (logits) | -39% | -2% |
> | sentence-level | (20, 11), **(18, 26)**, (14, 7), **(20, 20)**, **(14, 16)**, (14, 13), (22, 26), (28, 18),… | +10.5% (PPL) | -36% | -43% |
>
> The causal analysis reveals a 46.9% overlap (30 over 64, some examples of overlapped heads are marked in bold text) between the top-64 crucial heads for sentence- and word-level translation, indicating a shared core translation circuit. Ablating five shared heads severely degrades performance on both word-level (-39%) and sentence-level (-36%) tasks. Conversely, ablating five heads crucial only for sentence translation has a negligible impact on word-level performance (-2%) but substantially reduces the sentence-level performance (-43%).
>
> The behavioral pattern analysis of non-overlapping attention heads reveals their specialization in long-range dependencies and broader source contexts. Conversely, overlapping heads focus on local syntax and translation indicators.
>
> In summary, our analysis demonstrates the generalization of our method and findings to the sentence level. This provides a clearer mechanistic view of the model's translation process and underscores the broader applicability of our results.
>
> > Question #1 & 2: Some comments regarding the code
> >
>
> We thank the reviewer for their meticulous feedback on our code repository. We have addressed the points raised to improve the clarity and accessibility of our code:
>
> 1. Comment Language: To ensure our code is accessible to a wider audience, all comments have been rewritten in English.
> 2. Anonymity: We clarify that our implementation adapts public code from the "Interpretability in the Wild" (IOI) paper[6]. The reviewer-flagged comment (`# recall in IOI paper...`) is original to that codebase and was retained for attribution. The "we" in the comment, therefore, refers to the IOI authors[6], not the authors of this work, posing no risk to the blind review process.
>
> We can confirm there is **no author overlap**. To prevent any further misinterpretation, we have removed this specific comment and rephrased similar annotations inherited from the original codebase.
>
> > Question #3: Related work
> >
>
> We thank the reviewer for bringing this valuable work to our attention. We have now cited and discussed Zaranis et al. (2024)[7] in our Related Works section in the revised manuscript as follows:
>
> Zaranis et al. analyze context utilization in MT, examining how LLMs employ contextual elements like few-shot examples and source text during translation generation. This work analyzes translation from the view of input contexts, which is fundamentally distinct from our work, which focuses on mechanistic interpretability.
>
> > Question #4: Refinement of the title
> >
>
> Thank you very much for your thorough review. We will add the definite article as suggested to improve grammatical precision, revising the title to "Exploring the Translation Mechanism of Large Language Models.”
>
> > Reference:
> >
>
> [1] What the Harm? Quantifying the Tangible Impact of Gender Bias in Machine Translation with a Human-centered Study (EMNLP 2024)
>
> [2] Gender Bias in Large Language Models across Multiple Languages (TRUSTNLP 2025)
>
> [3] Interpreting and Improving Large Language Models in Arithmetic Calculation (ICML 2024)
>
> [4] Analyzing Multi-Head Self-Attention: Specialized Heads Do the Heavy Lifting, the Rest Can Be Pruned (ACL 2019)
>
> [5] How do Large Language Models Handle Multilingualism? (NIPS 2024)
>
> [6] Interpretability in the Wild: a Circuit for Indirect Object Identification in GPT-2 small (ICLR 2023)
>
> [7] Analyzing Context Contributions in LLM-based Machine Translation (EMNLP 2024)
>
> Thank you for your thorough review and helpful suggestions. We commit to incorporating the above analysis and experiments into our revision to address all these points.

---

> > ### Comment · Reviewer_LEwd · 2025-07-31
> > **Thank you for the additional experiments!**
> >
> > Thank you for the very elaborate explanations and results, it helped solidify my thoughts on the paper and its results. As I've already given an accept score (5) I'm keeping my score as is.

---

> ### Author Response · Authors · 2025-08-01
> **Sincerely appreciate for your valuable feedback!**
>
> Dear Reviewer LEwd,
>
> Thank you so much for your comprehensive and constructive review!
>
> All of your questions and comments were incredibly insightful and instrumental in improving the quality of our manuscript. The revisions we made based on your feedback have truly sharpened the core contributions and clarity of our work.
>
> We once again sincerely appreciate the valuable time and expertise you dedicated to our paper.
>
> With deepest thanks,
>
> All authors

---

### Official Review · Reviewer_QYtu · 2025-06-30

**Clarity:** 3
**Significance:** 3
**Originality:** 2
**Rating:** 3
**Confidence:** 4

**Summary:**

This paper systematically investigates the internal translation mechanisms within Large Language Models (LLMs), addressing the lack of fine-grained interpretability in multilingual translation tasks. The authors introduce a novel analysis framework called subspace-intervened path patching, enabling precise causal analysis of model components. Through this method, they identify a sparse subset (<5%) of attention heads critical for translation, categorized into source heads, indicator heads, and positional heads, whose outputs are integrated by Multi-Layer Perceptrons (MLPs) into English-centric latent representations. Empirical evaluations demonstrate that selectively fine-tuning these crucial components achieves translation performance comparable to full-parameter fine-tuning, with substantial parameter efficiency and generalization benefits.

**Questions:**

1. Generalization to Sentence-level Translation (Section 4.2, Line 181)
 You state that the identified components can generalize effectively to sentence-level translation. Is this generalization explicitly demonstrated using subspace-intervened path patching experiments at the sentence level, or is this claim solely supported by downstream fine-tuning tasks? If the former, please clarify the experimental setup and provide results; if the latter, explicitly state this limitation.
2. Clarification of Component Shifts after CPT vs. SFT (Line 221, Figures 1 and 3)
 In line 221, you mention significant distributional shifts in translation-crucial heads after continued pre-training (CPT), whereas minimal changes after supervised fine-tuning (SFT). However, Figures 1 and 3 do not clearly reflect this distinction. Could you provide additional analysis or quantitative measures (e.g., statistical comparisons, explicit annotations) to clearly illustrate and justify this observed difference?
Specifically, a direct visual or numerical comparison highlighting exactly which heads significantly shift and by how much would help clarify your claim.
3. Number and Variety of Prompts:
 You mentioned the prompt example "English: cloud - 中文: _" used in experiments. Did you utilize only this one prompt, or did you employ multiple prompts? Could you specify the exact number and variations of prompts used across your experiments?
4. Criteria for Defining Crucial Components:
 You indicated the importance of components by calculating relative changes and an importance score. Could you clarify explicitly how you determine the threshold to label a component as crucial? Provide precise numeric values or criteria employed to distinguish crucial from non-crucial components.
5. Statistical Significance of Behavioral Patterns Analysis:
 In Section 5 (Behavioral Patterns Analysis), you analyze attention heads and MLPs behaviors. Could you clarify the exact number of samples used in these analyses? Additionally, are these sample sizes sufficient to establish statistically significant patterns? Please provide statistical justifications or analyses that confirm the robustness of your observations.
6. Clarification of "Unembedding Matrix" (Line 277):
 Could you clearly define what you mean by the term "unembedding matrix"? Does it refer to the final projection matrix from hidden representations to vocabulary logits, or does it represent something else? Providing a formal or intuitive definition would enhance clarity.
7. Selection of Translation Indicator (IND), Source (SRC), and Target (TGT) Tokens:
 In the MLP analysis, how exactly do you select and categorize tokens into translation indicators (IND), source tokens (SRC), and target-language tokens (TGT)? Taking your example prompt explicitly ("English: cloud - 中文: _"), please clearly indicate which tokens belong to each category to eliminate ambiguity.

**Ethical Concerns:**

["NO or VERY MINOR ethics concerns only"]

**Final Justification:**

The overall framework and presentation are very similar to [1]. Even though the authors explained where they see the novelty, I don’t find the contribution particularly original. Also, the comparison with [1] is still not detailed enough—while the rebuttal added some experiments, they remain insufficient. Based on this, I will keep my original score.


Reference:
1. Interpreting and Improving Large Language Models in Arithmetic Calculation

**Limitations:**

yes

**Quality:**

3

**Strengths And Weaknesses:**

Strengths:
- Quality and Rigor:
 The methodological framework is well-developed and rigorously validated through comprehensive experimental setups, including various language pairs and multiple LLM architectures.
- Significance:
 Identifying a sparse set of critical components with demonstrated generalization across translation directions is significant for both interpretability and practical fine-tuning efficiency.
- Clarity:
 The paper is clearly structured, with thorough visualizations and detailed analyses that facilitate understanding of complex internal translation mechanisms.

Weaknesses:
- Novelty and Originality:
 The paper proposes an analytical approach (subspace-intervened path patching) that advances mechanistic interpretability specifically for LLM translation tasks, which is similar to the method used in [1].
- Scope Limitation:
 The analysis is currently limited to relatively simple, word-level translation tasks. Extending this framework to sentence-level or more realistic translation contexts would significantly strengthen its practical applicability.
- Open vs. Closed-source Limitation:
 The proposed interpretability method relies on the ability to access and intervene on internal model activations, limiting its direct application to closed-source or proprietary LLMs.
- Evaluation Breadth:
 Although thorough, evaluations primarily involve translation tasks across limited language pairs. Expanding the set of languages, especially including typologically diverse or low-resource languages, would help validate the robustness and universality of the proposed insights.

Reference:
1. Interpreting and Improving Large Language Models in Arithmetic Calculation

---

> ### Author Rebuttal · Authors · 2025-07-31
>
> Dear Reviewer QYtu,
>
> Thank you for your insightful reviews and comments. We appreciate the time and effort you have put into providing valuable feedback. We would like to address your concerns as follows:
>
> > W#1: Novelty and Originality
> >
>
> The proposed subspace-intervened path patching is distinct from the standard path patching method in [1].
>
> Standard path patching[1] intervenes on entire, full-dimensional activation vectors. In contrast, the proposed method intervenes on a task-specific, causally identified low-rank subspace within activations. In detail, by first identifying the task-relevant low-rank subspace responsible for the translation function (detailed in Algorithm 1) and then intervening only within that subspace via projection patching (Algorithm 2), we can isolate and study core translation mechanisms of the model, enabling a far more precise and fine-grained causal analysis.
>
> Moreover, the above novelty and originality are acknowledged by reviewer ByrS "By combining causal analysis with low-rank subspace decomposition, it proposes a novel path patching method that surpasses traditional activation patching." and reviewer NbyD "The paper introduces subspace-intervened path patching, which represents a methodological advancement over existing path patching techniques.".
>
> Furthermore, as we discuss in the response to Reviewer NbyD's Q#5, the applicability of the proposed method extends beyond translation, demonstrating its generalizability as a tool for mechanistic interpretability.
>
> > W#2: Scope Limitation
> >
>
> We wish to first clarify that the decision to focus initially on word-level translation was a deliberate methodological choice to isolate the core mechanisms of translation in a controlled environment, a common practice in previous, solid mechanistic interpretability research [1-3]. This avoids confounding variables at the sentence level—such as stylistic variations, non-trivial mapping, and paraphrasing that produce multiple valid translations—which obscure fine-grained causal analysis.
>
> As you suggested, we try to extend the analysis to the more complex task of sentence-level translation using the WMT23 En→Zh dataset, following the experimental procedures from Section 4.
>
> | En→Zh | Top Crucial Heads (Layer, Head) | Performance Metric Change (lower logits or higher PPL means poorer translation quality) | Performance Drop (Knockout Top-5 Overlapping Heads) | Performance Drop (Knock out Top-5 Sentence-Level Heads) |
> | --- | --- | --- | --- | --- |
> | word-level | (15, 21), (31, 11), **(18, 26)**, (16, 26), (31, 8), (26, 30), **(20, 20)**, **(14, 16)**,… | -4.47% (logits) | -39% | -2% |
> | sentence-level | (20, 11), **(18, 26)**, (14, 7), **(20, 20)**, **(14, 16)**, (14, 13), (22, 26), (28, 18),… | +10.5% (PPL) | -36% | -43% |
>
> The causal analysis reveals a 46.9% overlap (30 over 64, some examples of overlapped heads are marked in bold text) between the top-64 crucial heads for sentence- and word-level translation, indicating a shared core translation circuit. Ablating five shared heads severely degrades performance on both word-level (-39%) and sentence-level (-36%) tasks. Conversely, ablating five heads crucial only for sentence translation has a negligible impact on word-level performance (-2%) but substantially reduces the sentence-level performance (-43%).
>
> The behavioral pattern analysis of non-overlapping attention heads reveals their specialization in long-range dependencies and broader source contexts. Conversely, overlapping heads focus on local syntax and translation indicators.
>
> > W#3: Open vs. Closed-source Limitation
> >
>
> The study focuses on mechanistic interpretability, which necessitates reverse-engineering neural network computations. Consequently, accessing internal model components is a methodological prerequisite, not a limitation, aligning with standard practice in the field that primarily uses open-source models for foundational, reproducible discoveries [1-3]. Since current closed-source or proprietary LLMs share architectural homogeneity (i.e., the decoder-only Transformer), the proposed method can be directly used to understand these systems.
>
> > W#4: Evaluation Breadth
> >
>
> Thank you for your insightful comments, which inspire us a lot to strengthen this work.
>
> We wish to first clarify that our initial focus on high-resource language pairs was a deliberate methodological choice rather than an incidental weakness, enabling a clear and reliable analytical baseline. High translation accuracy is crucial for isolating and analyzing core translation mechanisms, ensuring findings are not confounded by the noise from inaccurate translations prevalent in low-resource settings.
>
> As suggested, we extend the analysis to a set of low-resource (sw, bn) and typologically diverse (bn, ar) language pairs. The results, shown in the following table, reveal that the key findings (the sparsity and transferability of key heads) hold across these low-resource and typologically diverse language pairs. These additional results confirm the universality and robustness of the findings.
>
> | Language Pair | Crucial Heads Proportion | Top Crucial Heads (Layer, Head) (overlapped marked in bold text) | Average Logits Change Ratio (the lower, the poorer translation quality) |
> | --- | --- | --- | --- |
> | En-Sw | 2.93% | **(16,26),(31,8),(18,11),(17,25)**,(15,17),… | -6.81% |
> | Zh-Sw | 3.32% | **(31,8),(18,11),(16,26),(17,25),(14,10)**,… | -7.19% |
> | En-Bn | 3.71% | **(30,18),(31,8),(14,10)**,(26,7),(28,20),… | -9.17% |
> | Zh-Bn | 2.34% | **(31,8),(30,18),(18,11),(14,10),(26,7)**,… | -8.20% |
> | En-Ar | 2.83% | (**30,18),(31,8),(14,10),(31,4)**,(20,18),… | -8.20% |
> | Zh-Ar | 2.05% | **(31,8),(30,18),(14,10),(31,4)**,(12,17),… | -8.94% |
>
> > Q#1: Generalization to Sentence-level Translation
> >
>
> The empirical downstream fine-tuning experiment supports this claim. We fine-tune only the crucial heads detected via subspace-intervened path patching to see whether it will improve the sentence-level translation performance compared to the random head fine-tuning. The result outperforms the random baseline and is competitive with Full SFT, supporting the generalization claim.
>
> To supplement the empirical result, we also conduct an analysis experiment via the mechanistic view at the sentence level to support the claim, please refer to the response to Weakness#1.
>
> > Q#2: Clarification of Component Shifts after CPT vs. SFT
> >
>
> To provide rigorous quantitative support for this observations, we analyzed the logits changes induced by SFT and CPT relative to the base model. We performed a Two-Sample Kolmogorov-Smirnov (K-S) test on the overall logits change distributions and also quantified the magnitude of change within the top 32 attention heads.
>
> | Comparison | K-S Test p-value | # Heads significantly shift (change > 1%) | Max logits change |
> | --- | --- | --- | --- |
> | Base vs. SFT | 0.355 | 8 of 32 | 3.12 |
> | Base vs. CPT | < 0.00001 | 17 of 32 | 12.03 |
>
> The results show that CPT induces a statistically significant distributional shift ($p<0.00001$), while SFT does not ($p=0.355$).
>
> > Q#3: Number and Variety of Prompts
> >
>
> We conducted experiments using 10 base prompts across 4 structural variation types, with details in Appendix B.2 and specific examples in Table 6.
>
> > Q#4: Criteria for Defining Crucial Components
> >
>
> We defined crucial components as those inducing a logits change of at least 1.0% (stated on Line 168). This threshold is grounded in both empirical analysis and established literature. Our causal analysis indicates that most attention head contributions fall within a ±1.0% range; therefore, this criterion effectively isolates components with a significant impact beyond baseline noise. Moreover, this data-driven threshold aligns with similar logit-based heuristics used in prior interpretability studies [4, 5].
>
> > Q#5: Statistical Significance of Behavioral Patterns Analysis
> >
>
> We utilized 100 randomly selected Zh↔En samples in both attention and MLP behavior analysis (stated on Line 266, 287), and it is sufficient to establish statistically significant patterns for two reasons:
>
> 1. Aligned with influential studies[2,4], the proposed interpretability approach prioritizes representative examples over quantity through manual inspection to uncover mechanistic behaviors.
> 2. As suggested, quantitative analysis of the key pattern (e.g., an attention head focusing on source tokens) shows 81 occurrences in 100 samples (81% consistency). The 95% Wilson score confidence interval [72.0%, 87.9%] exceeds chance (50%), indicating systematicity. A binomial test ($H₀: p=0.5$) rejected the null hypothesis ($p<0.001$), confirming significance.
>
> > Q#6: Clarification of "Unembedding Matrix"
> >
>
> The unembedding matrix, $W_{U} \in \mathbb{R}^{d_{\text{model}} \times |\mathcal{V}|}$, is the final linear layer that projects hidden states of dimension $d_{\text{model}}$ onto the vocabulary space of size $|\mathcal{V}|$.
>
> > Q#7: Elaboration of Specific Tokens
> >
>
> We use "English: cloud - 中文: 云" as an illustrative example:
>
> - IND: Instructional or structural tokens that frame the translation context but are not part of the source text. Here, IND tokens are 'English', ':', '-', and '中文'.
> - SRC: the input text for translation. Here, the SRC token is 'cloud'.
> - TGT: the translated output. Here, the TGT token is '云'.
>
> > Reference:
> >
>
> [1] Interpreting and Improving Large Language Models in Arithmetic Calculation (ICML 2024)
>
> [2] Analyzing Multi-Head Self-Attention: Specialized Heads Do the Heavy Lifting, the Rest Can Be Pruned (ACL 2019)
>
> [3] How do Large Language Models Handle Multilingualism? (NIPS 2024)
>
> [4] Interpretability in the Wild: a Circuit for Indirect Object Identification in GPT-2 Small (ICLR 2023)
>
> [5] How to use and interpret activation patching (Arxiv)
>
> Thank you for your thorough review and helpful suggestions. We commit to incorporating the above analysis and experiments into the revision.

---

> > ### Comment · Reviewer_QYtu · 2025-08-07
> >
> > Thanks for the rebuttal. I appreciate the effort, but I still feel the work mainly combines existing ideas (path patching + subspace) applied to a new task. The lack of comparison with prior path-patching methods also makes it hard to assess the novelty. I'll maintain the original score.

---

> ### Author Response · Authors · 2025-08-07
> **A Gentle Reminder for Reviewer QYtu**
>
> Dear Reviewer QYtu,
>
> We would be happy to discuss any further concerns you may have. Your insights are valuable to us, and we appreciate your time and attention to our work. We look forward to your feedback.
>
> Thank you very much.
>
> Best wishes,
>
> All Authors

---

> ### Author Response · Authors · 2025-08-08
> **Follow up on the discussion of the novelty of proposed method**
>
> Dear Reviewr QYtu,
>
> We thank you for the valuable comment and the opportunity to detail our method's novelty.
>
> First of all, we wish to clarify that the proposed method is more than a simple combination of prior techniques; it is a targeted and necessary design that addresses **polysemanticity** in LLMs—**a core challenge in mechanistic interpretability** [2-4]—to enable a fine-grained exploration of the model's translation mechanism (the primary contribution of this work).
>
> **Conceptual Novelty: From Full-Vector to Subspace Intervention**
>
> Standard path patching [1] intervenes on the **entire activation vector** of a component. However, these activations are often **polysemantic**—they simultaneously encode multiple, unrelated concepts [2-3]. Consequently, full-vector patching conflates the causal effects of a target function (e.g., translation) with numerous irrelevant functions encoded in the same vector, failing to achiveve our objectives.
>
> Our method addresses this by identifying the low-dimensional subspace **specifically responsible for translation** within the activation space and intervening *only* on this subspace. It allows us to isolate the specific causal mechanism of translation from confounding functionalities, providing a more fine-grained and accurate understanding of the model's internal translation mechanism. This core novelty and contribution, facilitated by our novel method, has been acknowledged by Reviewers Byrs, Nbyd, and LEwd.
>
> **Empirical validation confirms this advancement**
>
> We did our best to reproduce standard path patching baseline and conduct comparable experiments; The empirical results demonstrate our fine-grained method's superiority over this baseline [1]. The table below compares both approaches across high- and low-resource translation directions.
>
> | **Translation Pairs (standard indicates method in [1] and subspace-intervened is ours)** | **Top Crucial Heads (Layer, Head)** | **Avg. Logits Change** | **Acc. Drop (Knockout Top-5)** | **Targeted SFT Performance (BLEU/COMET/BLEURT)** |
> | --- | --- | --- | --- | --- |
> | En→Zh (standard) | (31, 8), (14, 10), (30, 18), (12, 17), (16, 26), (15, 9), (31, 4), (22, 6),… | -2.69% | -25% | 27.3/79.8/62.4 |
> | En→Zh (subspace-intervened) | (15, 21), (31, 11), (18, 26), (16, 26), (31, 8), (26, 30), (20, 20), (14, 16),… | **-4.47%** | **-39%** | **28.9/80.5/63.1** |
> | Zh→En (standard) | (15, 19), (31, 22), (14, 10), (30, 18), (22, 10), (15, 11), (31, 4), (14, 14),… | -1.71% | -22% | 18.5/77.9/62.8 |
> | Zh→En (subspace-intervened) | (31, 27), (31, 11), (14, 14), (15, 19), (31, 4), (26, 30), (14, 7), (30, 12),… | **-2.49%** | **-31%** | **19.8/78.4/63.3** |
> | En→Sw (standard) | (22, 17), (31, 8), (16, 6), (20, 14), (14, 10), (30, 26), (16, 10), (12, 17),… | -3.12% | -28% | 1.83/51.5/40.9 |
> | En→Sw (subspace-intervened) | (16, 26), (31, 8), (18, 11), (17, 25), (15, 17), (14, 10), (30, 12), (20, 14),… | **-6.81%** | **-42%** | **3.91/55.1/43.7** |
> | Sw→En (standard) | (14, 14), (31, 22), (15, 11), (18, 26), (14, 7), (22, 26), (15, 19), (17, 18),… | -1.43% | -21% | 14.5/67.1/53.2 |
> | Sw→En (subspace-intervened) | (31, 27), (30, 18), (14, 10), (31, 22), (15, 19), (14, 14), (31, 4), (18, 26),… | **-2.01%** | **-26%** | **15.9/67.9/54.0** |
>
> Results show our method identifies components more critical to translation, as intervening on the heads identified by our method produces a larger drop in translation quality across all directions than the standard path patching, further confirmed by knockout validation. Moreover, targeted fine-tuning only the top-32 heads identified by our method (subspace-intervened) yields superior translation performance in all directions, enabling more targeted enhancement compared to the standard one.
>
> To sum up, as the above theoretical clarification and empirical results present, the proposed subspace-intervened path patching, which distinctly differs from a simple combination of prior techniques, is specifically designed to enable the **finer-grained, targeted** mechanistic analysis in polysemantic LLMs for mechanistic interpretability, facilitating the core contribution of the accurate and specific exploration of translation mechanism.
>
> ---
>
> Reference:
>
> [1] Interpreting and Improving Large Language Models in Arithmetic Calculation
>
> [2] Elhage, et al., "Toy Models of Superposition", Transformer Circuits Thread, 2022.
>
> [3] Polysemanticity and Capacity in Neural Networks
>
> [4] What Causes Polysemanticity? An Alternative Origin Story of Mixed Selectivity from Incidental Causes

---

> > ### Author Response · Authors · 2025-08-09
> > **Looking forward to your valuable feedback!**
> >
> > Dear Reviewer QYtu,
> >
> > In the previous response, we have clarified in detail that our approach's novelty lies in its specific design for fine-grained mechanistic interpretability analysis, rather than a simple combination of existing methods. As the discussion period concludes, we look forward to further discussion.
> >
> > Thank you once again for your thoughtful review and support.
> >
> > Best regards, Authors

---

### Official Review · Reviewer_NbyD · 2025-06-30

**Clarity:** 2
**Significance:** 2
**Originality:** 2
**Rating:** 5
**Confidence:** 3

**Summary:**

This paper presents a framework for analyzing the internal translation mechanisms of large language models (LLMs), with a focus on understanding how these models perform multilingual translation at the computational component level. The work addresses a gap in mechanistic interpretability by moving beyond surface-level observations to analyze the causal relationships between model components and translation capability. The authors propose "subspace-intervened path patching," a refinement of existing path patching techniques that enables more precise causal analysis by identifying and intervening only within task-specific "translation-steering" subspaces of component activations, rather than manipulating entire activation vectors. Through systematic application of their method, the authors identify that translation is driven by a remarkably sparse subset of components (less than 5% of attention heads). The authors demonstrate that targeted fine-tuning of only the identified crucial components (64 attention heads, <5% of parameters) achieves performance parity with full-parameter fine-tuning.

**Questions:**

1. You analyze word-level translation but claim generalization to sentence level based on fine-tuning results. Can you extend your mechanistic analysis to phrases and sentences?
2. Can you elaborate on why fine-tuning only 64 heads is sufficient for strong translation performance?
3. You show that targeted fine-tuning achieves similar performance to full fine-tuning, but can you provide more detailed analysis of the trade-offs? Are there specific translation phenomena where targeted fine-tuning consistently underperforms? What's the performance ceiling of this approach?
4. Do the translation-crucial components emerge from multilingual pre-training data exposure, or do they develop during fine-tuning?
5. Could your subspace analysis be applied to other tasks beyond translation?

**Ethical Concerns:**

["NO or VERY MINOR ethics concerns only"]

**Final Justification:**

The authors addressed some of my concerns in the rebuttal. I have adjusted the final score accordingly.

**Limitations:**

yes

**Quality:**

2

**Strengths And Weaknesses:**

### Strengths:
- The paper introduces subspace-intervened path patching, which represents a methodological advancement over existing path patching techniques.
- The research questions framework (Which components? What behavioral patterns? Can we improve?) provides good organization.

### Weaknesses:
- The paper focuses exclusively on word-level translation, which may not capture the complexity of real-world translation scenarios. While the authors claim generalization to sentence-level translation based on fine-tuning results, the mechanistic analysis itself is limited to single-word mappings.
- The discovery of English-centric intermediate representations is an important insight, but the current analysis is limited to cosine similarity of MLP outputs with English token embeddings, which is only a coarse signal. The paper does not investigate why or how English emerges as a pivot in non-English ↔ non-English translation pairs.

---

> ### Author Rebuttal · Authors · 2025-07-31
>
> Dear Reviewer NbyD,
>
> Thank you for your insightful reviews and comments. We appreciate the time and effort you have put into providing valuable feedback. We would like to address your concerns as follows:
>
> > W #1: Mechanistic analysis is limited to word-level translation
> >
>
> We wish to first clarify that the decision to focus initially on word-level translation was a deliberate methodological choice to isolate the core mechanisms of translation in a controlled environment, a common practice in previous, solid mechanistic interpretability research [1-3]. This avoids confounding variables at the sentence level—such as stylistic variations, non-trivial mapping, and paraphrasing that produce multiple valid translations—which obscure fine-grained causal analysis.
>
> As you suggested, we try to extend the analysis to the more complex task of sentence-level translation using the WMT23 En→Zh dataset, following the experimental procedures from Section 4.
>
> | En→Zh | Top Crucial Heads (Layer, Head) | Performance Metric Change (lower logits or higher PPL means poorer translation quality) | Performance Drop (Knockout Top-5 Overlapping Heads) | Performance Drop (Knock out Top-5 Sentence-Level Heads) |
> | --- | --- | --- | --- | --- |
> | word-level | (15, 21), (31, 11), **(18, 26)**, (16, 26), (31, 8), (26, 30), **(20, 20)**, **(14, 16)**,… | -4.47% (logits) | -39% | -2% |
> | sentence-level | (20, 11), **(18, 26)**, (14, 7), **(20, 20)**, **(14, 16)**, (14, 13), (22, 26), (28, 18),… | +10.5% (PPL) | -36% | -43% |
>
> The causal analysis reveals a 46.9% overlap (30 over 64, some examples of overlapped heads are marked in bold text) between the top-64 crucial heads for sentence- and word-level translation, indicating a shared core translation circuit. Ablating five shared heads severely degrades performance on both word-level (-39%) and sentence-level (-36%) tasks. Conversely, ablating five heads crucial only for sentence translation has a negligible impact on word-level performance (-2%) but substantially reduces the sentence-level performance (-43%).
>
> The behavioral pattern analysis of non-overlapping attention heads reveals their specialization in long-range dependencies and broader source contexts. Conversely, overlapping heads focus on local syntax and translation indicators.
>
> > W #2: Coarse analysis of English-centric intermediate representations
> >
>
> Our paper's primary objective was to first identify and characterize this phenomenon. The use of cosine similarity is a direct and established method for this characterization as used in prior research [4-6].
>
> We then conducted a new correlation analysis to demonstrate that this English-centricity has a direct and significant impact on translation performance. We measured the Pearson correlation between the representation's cosine similarity to English and the final translation quality (BLEU, chrF, TER) for 12 resource levels and typologically diverse non-English to non-English language pairs. The results shown in the following table reveal a strong correlation and provide compelling evidence that the English-centric intermediate representation is not a superficial phenomenon but a core factor fundamentally linked to translation quality.
>
> | Correlation with English Similarity | BLEU-1 Score | chrF Score | TER Score |
> | --- | --- | --- | --- |
> | Average across 12 language pairs | 0.905 | 0.873 | -0.919 |
>
> To investigate **why** English emerges as the pivot, we hypothesized that the pivot language corresponds to the dominant language in the model's pre-training corpus. The Llama models analyzed in our paper were pre-trained on a corpus where English is overwhelmingly dominant [7]. In addition, we conduct a parallel experiment on the Qwen2.5, a model known to be pre-trained on a corpus with a predominance of Chinese data[8]. In this context, we found that Chinese, not English, emerged as the pivot language. This preliminary analysis result somehow supports our hypothesis.
>
> To illustrate **how** this pivot emerges mechanistically within the model, we performed a logits lens analysis aligned with [4-5]. This visualization reveals a progressive transition: representations of source concepts shift towards their English counterparts in the model's intermediate layers before being translated into the final target language. For instance, when translating “车” (Chinese for *car*) to “voiture” (French), the representation explicitly resolves to “car” around layers 19-27 before shifting to “voiture” in the final layers.
>
> > Q #1: Extend mechanistic analysis to sentences
> >
>
> For the extended mechanistic analysis of sentence-level translation, please refer to the response to W #1.
>
> > Q #2: Why fine-tunig 64 heads achieve strong performance
> >
>
> Our claim that fine-tuning only 64 heads is sufficient for strong translation performance is principled, grounded in causal analysis and empirical validation:
>
> 1. Our causal analysis identifies that a small subset of attention heads (64 of 1024) dominantly contributes to translation, accounting for over 80% of the significant logit modifications.
> 2. Our ablation studies (Table 3) reveal that fine-tuning 64 heads offers an optimal trade-off, achieving performance comparable to Full SFT at a substantially lower computational cost, as further increases in trainable heads yield diminishing returns.
>
> > Q #3: Concerns regarding targeted SFT
> >
>
> Tables 3 and 4 quantitatively detail the trade-offs between translation performance, training efficiency, computational cost, and catastrophic forgetting. Increasing fine-tuned heads incrementally improves performance but proportionally raises memory consumption and training time. Crucially, aggressively tuning excessive heads exacerbates catastrophic forgetting, degrading general capabilities.
>
> Error analysis of underperforming case (Zh→En) clusters revealed the top-three error patterns accounting for >70% of significant performance gaps:
>
> - Style/Diction/Idioms: Targeted SFT yields overly literal translations (e.g., "*drug market*" vs. correct idiomatic "pharmaceutical showcase" for`新冠肺炎对毒品市场的影响`(COVID-19's impacts on the pharmaceutical showcase)).
> - Noisy Data Robustness: Reduced resilience to ambiguous inputs (e.g., misinterpreting `第9草`(Article 9) as "*9th draft*" vs. the correct "Article 9").
> - Factual Hallucinations: Generating unsupported details (e.g., adding "*green light*" to `充电盒未充满电充电指示灯红灯长亮...`(The charging indication light on the charging box is not yet fully charged, the red light is on)).
>
> Our method achieves performance ceiling statistically comparable to full fine-tuning by exclusively tuning the 64 heads most critical for translation with lower computational cost, as demonstrated empirically in Tables 3 and 4.
>
> > Q #4: Discussion of the emergence of translation-crucial components
> >
>
> As Section 4.4 claimed, translation-crucial components are formed during multilingual pre-training and are subsequently refined by SFT. The following comparative causal analysis supports this claim.
>
> 1. A randomly initialized model exhibits an unstructured logit change matrix with no evidence of specialized translation heads. In contrast, the pre-trained LLaMA-2 model develops a sparse set of critical translation heads, representing a statistically significant distributional shift from the random baseline ($p<0.05$, Figure 1). This demonstrates that these functional circuits emerge during the pre-training phase.
> 2. The subsequent transition from the pre-trained to the fine-tuned model induces only a minor, statistically insignificant distributional shift ($p>0.05$, Figure 3). This indicates that SFT primarily enhances or slightly adjusts these pre-existing components rather than forming them.
>
> > Q #5: Can subspace analysis be applied to other tasks
> >
>
> Our method is task-agnostic, adaptable to new tasks by constructing task-specific analysis datasets. To demonstrate this versatility, we applied our analysis to multilingual mathematical reasoning, creating counterfactual examples from the MGSM dataset [9]. Following the principles detailed in Section 3, we consider the following examples from our multilingual mathematical reasoning analysis:
>
> - X_f: "肖恩有五个玩具。圣诞节他从他爸爸妈妈那里各得到了两个玩具。他现在有多少个玩具？请给出数字: " ("Shawn has five toys. For Christmas, he got two toys each from his mom and dad. How many toys he has now? Give the number: ")
> - X_cf: "肖恩有五个玩具。圣诞节他从他爸爸妈妈那里各得到了两个玩具。他现在有多少个玩具？请转述句子: " ("Shawn has five toys. For Christmas, he got two toys each from his mom and dad. How many toys he has now? Rephrase the sentence: ")
>
> The analysis identified a sparse set of crucial heads for this new task (3.95% among all heads), mirroring the trend observed in translation. The top-5 heads (e.g., (11, 8), (12, 22), (6, 22), (18, 12), (4, 31)) induced an average logit decrease of 9.76%. Subsequently, ablating the top-10 heads caused a substantial performance drop of approximately 60% in task accuracy. These results confirm that our method effectively identifies components critical to mathematical reasoning, validating its task-agnostic nature.
>
> > Reference:
> >
>
> [1] Interpreting and Improving Large Language Models in Arithmetic Calculation (ICML 2024)
>
> [2] Analyzing Multi-Head Self-Attention: Specialized Heads Do the Heavy Lifting, the Rest Can Be Pruned (ACL 2019)
>
> [3] How do Large Language Models Handle Multilingualism? (NIPS 2024)
>
> [4] Do Llamas Work in English? On the Latent Language of Multilingual Transformers (ACL 2024)
>
> [5] How do Large Language Models Handle Multilingualism? (NIPS 2024)
>
> [6] Transformer Feed-Forward Layers Build Predictions by Promoting Concepts in the Vocabulary Space (EMNLP 2022)
>
> [7] Llama 2: Open Foundation and Fine-Tuned Chat Models (Arxiv)
>
> [8] Qwen2.5 Technical Report (Arxiv)
>
> [9] Language Models are Multilingual Chain-of-Thought Reasoners (ICLR 2023)
>
> Thank you for your thorough review and helpful suggestions. We commit to incorporating the above analysis and experiments into the revision to address all these points.

---

> > ### Author Response · Authors · 2025-08-04
> > **Follow-up on Our Rebuttal Submission**
> >
> > Dear Reviewer NbyD,
> >
> > I'm writing to express our gratitude for the time and effort you've dedicated to reviewing our paper. We have addressed the points you raised in detail in our responses.
> >
> > As the discussion period is coming to a close soon, we kindly ask if you could review our responses at your earliest convenience. We are eager to know if our explanations have alleviated your concerns. If there are still areas needing improvement, your insights would be greatly appreciated and instrumental in enhancing our work.
> >
> > Thank you once again for your thoughtful review and support.
> >
> > Warm regards, Authors

---

> > ### Comment · Reviewer_NbyD · 2025-08-09
> >
> > Thanks for answering my questions. I will adjust my score accordingly. All the best.

---

> ### Author Response · Authors · 2025-08-08
> **A Gentle Reminder for Reviewer NbyD**
>
> Dear Reviewer NbyD,
>
> We are writing to kindly remind you that the discussion period will soon come to a close. We understand that you may be very busy, and we are worried that we may miss the time to respond to your feedback. We look forward to receiving your valuable feedback and having the opportunity to address your further concerns.
>
> Thank you for your time and effort.
>
> Best regards,
> All Authors

---

> > ### Author Response · Authors · 2025-08-09
> > **Looking forward to your feedback!**
> >
> > Dear Reviewer NbyD,
> >
> > As the discussion period is coming to a close in 12 hours, we kindly ask if you could review our responses at your earliest convenience.
> >
> > Thank you for your time and effort.
> >
> > Best regards, All Authors

---

### Official Review · Reviewer_ByrS · 2025-07-01

**Clarity:** 4
**Significance:** 4
**Originality:** 4
**Rating:** 6
**Confidence:** 5

**Summary:**

The paper reveals the sparsity and English-centric bridging mechanism in LLM translation through subspace intervention and component-level analysis. Based on this discovery, it proposes an efficient targeted fine-tuning method that achieves full-parameter fine-tuning performance by optimizing only 64 attention heads while doubling training speed, laying the foundation for interpretable machine translation research.

**Questions:**

1. Unsupported percentage claims - the 60% and 70% overlap rates lack clear evidence in Figure 1.
2. Figure 1 contains six subfigures, each showing position (8,31), but no direct comparison demonstrates the claimed logit decrease when patching this head, and red/brown and gray/purple are difficult to distinguish.
3. The claimed 70.0% impact in MLP detection results isn't visually supported in Figure 1 - color changes from white to dark only show ~50%+ difference.
4. Figure 2 lacks a baseline reference line, only showing accuracy differences when removing different key heads and key MLPs.

**Ethical Concerns:**

["NO or VERY MINOR ethics concerns only"]

**Final Justification:**

Thank you for the detailed response and additional experiments. I confirm that all my concerns have been fully addressed.

**Limitations:**

The potential bias amplification risks associated with fine-tuning critical components (e.g., whether the English-centric bridging mechanism reinforces linguistic hegemony) were not discussed. It is recommended to supplement this analysis.

**Paper Formatting Concerns:**

No serious formatting issues were found.

**Quality:**

4

**Strengths And Weaknesses:**

Quality：​
The proposed Subspace-Intervened Path Patching technique precisely identifies translation-critical components by comparing activation differences between positive and negative samples. The experimental design covers multiple language pairs (e.g., Chinese-English, German-French) and model scales (LLaMA2-7B/13B, Mistral-7B), ensuring high credibility of results.

Clarity：
The subspace decomposition and path patching process are clearly described, enhanced by visualizations for better understanding.

Significance：​​
It systematically uncovers the sparsity mechanism and English-centric bridging pattern in LLM translation for the first time, filling a gap in LLM interpretability research.

Originality：​
By combining causal analysis with low-rank subspace decomposition, it proposes a novel path patching method that surpasses traditional activation patching.

Weakness：
The study does not explore the implications of its findings for domain-adaptive translation (e.g., medical, legal).
It does not evaluate the long-term stability of models after targeted fine-tuning.

---

> ### Author Rebuttal · Authors · 2025-07-31
>
> Dear Reviewer ByrS,
>
> Thank you for your insightful reviews and comments. We appreciate the time and effort you have put into providing valuable feedback. We would like to address your concerns as follows:
>
> > Weakness #1: Limited evaluation on domain-adaptive translation
> >
>
> We thank the reviewer for their insightful feedback regarding broader applicability and long-term implications.
>
> Our primary focus is general-domain translation, and we acknowledge the importance of specialized domains. We therefore conducted new experiments using medical (ELRC-Medical-V2[1], En→De) and legal (M3T[2], En→Zh) benchmarks:
>
> | Lang Pair | Domain | Random SFT (BLEU/COMET/BLEURT) | Targeted SFT (BLEU/COMET/BLEURT) | Full SFT (BLEU/COMET/BLEURT) |
> | --- | --- | --- | --- | --- |
> | En → De | Medical | 28.9/83.9/73.8 | 39.9/87.4/77.5 | 41.0/88.5/79.1 |
> | En → Zh | Legal | 8.07/75.2/65.8 | 45.8/89.2/78.1 | 52.2/90.5/80.5 |
>
> Analysis shows Targeted SFT remains highly competitive—significantly outperforming random baselines—but doesn’t match Full SFT in specialized domains. We attribute this gap to:
>
> 1. Since the training and test sets share the same distribution (via a split of one dataset), Full SFT is prone to overfitting, but Targeted SFT is not.
> 2. Domain-Specific Patterns: Unique syntax and low-frequency terminology may require modifying more parameters than our targeted approach adjusts.
> 3. Head Specialization: Attention heads optimized for general-domain translation may not fully overlap with those essential for specialized domains.
> This reveals a trade-off between parameter efficiency and peak performance in specialized domains, now discussed in Section 6 as future work.
>
> We appreciate these constructive comments, which strengthened our work. This point is added to our discussion for future investigation.
>
> > Question #1: Unsupported percentage claims
> >
>
> To quantitatively address your concerns regarding the percentage claims, we provide detailed statistics in the following paragraphs. The following numeric results show that while some language pairs have moderate overlap (e.g., Chinese-English/English-Chinese at 40.62%), others exhibit significantly higher rates. Specifically, the overlap rates are:
>
> - 40.62% between Chinese-English (`zh-en`) and English-Chinese (`en-zh`).
> - 62.50% between Chinese-French (`zh-fr`) and French-Chinese (`fr-zh`).
> - 75.00% between English-Chinese (`en-zh`) and French-Chinese (`fr-zh`).
>
> The original statement intended to convey that overlap rates for many language-direction pairs could reach as high as 70%. We acknowledge that this was not articulated with sufficient precision. We have corrected the text to explicitly state the specific pairs corresponding to these higher percentages, thereby removing ambiguity and directly linking our claim to the evidence in Figure 1.
>
> > Question #2: Elaboration on concerns regarding Figure 1
> >
>
> We thank the reviewer for their careful attention to Figure 1, and we wish to clarify that Figure 1 is designed to provide a direct comparison. The color intensity of each square visually represents the magnitude of the logit change resulting from patching the corresponding attention head—a deeper red indicates a more significant logit decrease.
>
> The consistent deep red of the square at position (8,31) across all six subfigures demonstrates its critical negative impact on performance in all tested translation directions. To supplement this visual data, we provide the specific quantitative values for the average logit decrease when patching head (8,31):
>
> - Zh → En: -1.70
> - Zh → Fr: -2.80
> - Zh → Ru: -1.20
> - En → Zh: -1.10
> - Fr → Zh: -3.20
> - Ru → Zh: -5.00
>
> These results confirm that patching the head (8,31) consistently and substantially degrades model performance.
>
> To clearly distinguish the results from heads and MLPs, we adopted a color scheme consisting of blue, orange, sky blue, vermilion, bluish green, yellow, and reddish purple to ensure all values are easily distinguishable. We appreciate your thorough review which greatly improve the clarity and presentation of our paper.
>
> > Question #3: Unsupported percentage claims of MLP
> >
>
> We thank the reviewer for their careful observation and for the opportunity to provide clarification.
>
> While we agree that a visual inspection of the heatmap might suggest an average impact closer to 50% in some language directions, our claim of reaching 70.0% is based on the maximum effect observed in our data in several language directions, not the average. The heatmap serves as a qualitative illustration of the overall trend, whereas the specific claims in the text are supported by precise quantitative results.
>
> To make this explicit, the exact impact values for the key language pairs in Figure 1 are:
>
> - French → Chinese (`Fr → Zh`): −70.83%
> - English → Chinese (`En → Zh`): −66.87%
> - Russian → Chinese (`Ru → Zh`): −56.64%
> - Chinese → Russian (`Zh → Ru`): −55.79%
> - Chinese → English (`Zh → En`): −52.66%
> - Chinese → French (`Zh → Fr`): −50.07%
>
> As the data shows, the impact for the `Fr → Zh` pair is indeed over 70.0%. To prevent any future ambiguity, we have revised the manuscript to explicitly state this peak numerical value when referencing Figure 1, ensuring the claim is directly and unmistakably supported.
>
> > Question #4: Elaboration on concerns regarding Figure 2
> >
>
> We thank the reviewer for the opportunity to clarify the experimental design shown in Figure 2. The performance degradation from removing *randomly selected* heads and MLPs is the intended baseline for this analysis. The figure's purpose is to demonstrate that the components identified by our method are significantly more impactful than arbitrarily chosen ones. Therefore, this "random removal" condition provides the most direct and rigorous baseline for validating our method's precision.
>
> An obvious note specifying the baseline has been included in the revised manuscript to provide a clearer and comprehensive comparison of the experimental results.
>
> > Limitation #1: Supplemental analysis of potential bias amplification
> >
>
> We thank the reviewer for this insightful comment. To empirically address the potential risk of linguistic hegemony, we have conducted a dedicated analysis to evaluate whether our targeted fine-tuning approach amplifies translation biases.
>
> We assessed our method using the CSI-Match metric [3] on the CAMT dataset [3]. CSI-Match is designed to measure the translation accuracy of culturally specific terms, where higher scores indicate a lower risk of linguistic hegemony [3, 4]. We compared our Targeted SFT against the Base model, Full SFT, and Random SFT baselines.
>
> | Model (En→Zh) | BLEU | COMET | BLEURT | CSI-Match |
> | --- | --- | --- | --- | --- |
> | Base (Llama-2-7B) | 19.54 | 73.57 | 51.02 | 16.12 |
> | w/ Full SFT | 25.50 | 79.35 | 58.28 | 18.44 |
> | w/ Targeted SFT | 25.85 | 79.58 | 58.64 | 18.62 |
> | w/ Random SFT | 19.98 | 74.73 | 52.88 | 16.13 |
>
> The results show that our Targeted SFT achieves a CSI-Match score comparable to the more resource-intensive Full SFT baseline, with no statistically significant difference between them. This finding provides strong evidence that our targeted approach successfully improves performance without introducing additional bias amplification risks compared to standard full fine-tuning.
>
> We have incorporated this analysis into the revised manuscript. We thank the reviewer for this valuable suggestion, which has strengthened our paper.
>
> > Reference:
> >
>
> [1] European Language Resource Coordination: Collecting Language Resources for Public Sector Multilingual Information Management (LREC 2018)
>
> [2] M3T: A New Benchmark Dataset for Multi-Modal Document-Level Machine Translation (NAACL 2024)
>
> [3] Benchmarking Machine Translation with Cultural Awareness (EMNLP 2024)
>
> [4] Towards Cross-Cultural Machine Translation with Retrieval-Augmented Generation from Multilingual Knowledge Graphs (EMNLP 2024)
>
> Thank you for your thorough review and helpful suggestions. We commit to incorporating the above analysis and experiments into the revision to address all these points.

---

> > ### Comment · Reviewer_ByrS · 2025-08-05
> >
> > Thank you for the detailed response and additional experiments. I confirm that all my concerns have been fully addressed.

---

### Note · Authors · 2025-08-16

Dear SAC, AC, and Reviewers,

We are sincerely grateful for the time and effort you have all dedicated to reviewing our manuscript. Reviewers' thoughtful feedback and constructive comments have been invaluable in helping us to significantly improve the quality of our work.

As the discussion period concludes, we wish to provide a brief summary to assist in the final decision-making process. We have diligently worked to address all points raised and have provided a comprehensive and active response to the further concerns from Reviewer QYtu, and we are still not receiving any follow-up from Reviewer QYtu. We also respectfully note that the final ratings from two reviewers are still pending.

During the rebuttal period, we conducted additional experiments to thoroughly address the primary concerns raised. Specifically, we have expanded our causal analysis to the sentence-level, broadened the evaluation to include a wider range of language pairs, and addressed all other points mentioned in the reviews. We are confident that these revisions and clarifications have successfully resolved the reviewers' concerns and have further strengthened the contributions of our paper.

We hope this information is helpful for your final assessment. Thank you once again for your guidance and for considering our work.

Sincerely,

All Authors

---

### Decision · Program_Chairs · 2025-09-17

**Decision:**

Accept (poster)

**Comment:**

This paper proposes a mechanistic interpretability study of the internal translation mechanisms in LLMs to shed light on fine-grained interpretability in multilingual translation tasks. The main contribution is a framework called "subspace-intervened path patching" which extends standard path patching. A sparse subset (<5%) of attention heads is identified which is critical for translation, whose outputs are integrated by Multi-Layer Perceptrons (MLPs) into English-centric latent representations. Empirical evaluations demonstrate that selectively fine-tuning these crucial components achieves translation performance comparable to full-parameter fine-tuning. The main weaknesses pointed out by reviewers are the limited novelty of the  path patching approach over previous existing ones, the analysis being limited to word-level translation tasks (rather than sentence, paragraph, or document-level translation), and the limited set of languages considered. In their rebuttal, the authors convincingly addressed all these points, clarifying the differences and significance of their proposed approach with respect to standard path patching, providing new sentence-level experiments and adding new languages. I therefore recommend acceptance.